# Chitinase 3-like-1 contributes to acetaminophen-induced liver injury by promoting hepatic platelet recruitment

Zhao Shan[1,2], Leike Li[3], Constance Lynn Atkins[1], Meng Wang[1], Yankai Wen[1], Jongmin Jeong[1], Nicolas F Moreno[1], Dechun Feng[4], Xun Gui[3], Ningyan Zhang[3], Chun Geun Lee[5], Jack A Elias[5,6], William M Lee[7], Bin Gao[4], Fong Wilson Lam[8,9], Zhiqiang An[3]*, Cynthia Ju[1]*

[1]Department of Anesthesiology, UTHealth McGovern Medical School, Houston, United States; [2]Center for Life Sciences, School of Life Sciences, Yunnan University, Kunming, China; [3]Texas Therapeutics Institute, UTHealth McGovern Medical School, Houston, United States; [4]Laboratory of Liver Disease, National Institute on Alcohol Abuse and Alcoholism, NIH, Bethesda, United States; [5]Molecular Microbiology and Immunology, Brown University, Providence, United States; [6]Division of Medicine and Biological Sciences, Warren Alpert School of Medicine, Brown University, Providence, United States; [7]Division of Digestive and Liver Diseases, Department of Internal Medicine, University of Texas Southwestern Med School, Dallas, United States; [8]Division of Pediatric Critical Care Medicine, Baylor College of Medicine, Houston, United States; [9]Center for Translation Research on Inflammatory Diseases, Michael E. DeBakey Veterans Affairs Medical Center, Houston, United States

*For correspondence:
zhiqiang.an@uth.tmc.edu (ZA);
Changqing.Ju@uth.tmc.edu (CJ)

Competing interests: The authors declare that no competing interests exist.

## Abstract

**Background:** Hepatic platelet accumulation contributes to acetaminophen (APAP)-induced liver injury (AILI). However, little is known about the molecular pathways involved in platelet recruitment to the liver and whether targeting such pathways could attenuate AILI.

**Methods:** Mice were fasted overnight before intraperitoneally (*i.p.*) injected with APAP at a dose of 210 mg/kg for male mice and 325 mg/kg for female mice. Platelets adherent to Kupffer cells were determined in both mice and patients overdosed with APAP. The impact of α-chitinase 3-like-1 (α-Chi3l1) on alleviation of AILI was determined in a therapeutic setting, and liver injury was analyzed.

**Results:** The present study unveiled a critical role of Chi3l1 in hepatic platelet recruitment during AILI. Increased Chi3l1 and platelets in the liver were observed in patients and mice overdosed with APAP. Compared to wild-type (WT) mice, *Chil1*[-/-] mice developed attenuated AILI with markedly reduced hepatic platelet accumulation. Mechanistic studies revealed that Chi3l1 signaled through CD44 on macrophages to induce podoplanin expression, which mediated platelet recruitment through C-type lectin-like receptor 2. Moreover, APAP treatment of *Cd44*[-/-] mice resulted in much lower numbers of hepatic platelets and liver injury than WT mice, a phenotype similar to that in *Chil1*[-/-] mice. Recombinant Chi3l1 could restore hepatic platelet accumulation and AILI in *Chil1*[-/-] mice, but not in *Cd44*[-/-] mice. Importantly, we generated anti-Chi3l1 monoclonal antibodies and demonstrated that they could effectively inhibit hepatic platelet accumulation and AILI.

**Conclusions:** We uncovered the Chi3l1/CD44 axis as a critical pathway mediating APAP-induced hepatic platelet recruitment and tissue injury. We demonstrated the feasibility and potential of targeting Chi3l1 to treat AILI.

**Funding:** ZS received funding from NSFC (32071129). FWL received funding from NIH (GM123261). ALFSG received funding from NIDDK (DK 058369). ZA received funding from CPRIT (RP150551 and RP190561) and the Welch Foundation (AU-0042–20030616). CJ received funding from NIH

(DK122708, DK109574, DK121330, and DK122796) and support from a University of Texas System Translational STARs award. Portions of this work were supported with resources and the use of facilities of the Michael E. DeBakey VA Medical Center and funding from Department of Veterans Affairs I01 BX002551 (Equipment, Personnel, Supplies). The contents do not represent the views of the US Department of Veterans Affairs or the US Government.

## Introduction

Acute liver failure (ALF) is a life-threatening condition of massive hepatocyte injury and severe liver dysfunction that can result in multi-organ failure and death (*Bernal and Wendon, 2013*). Acetaminophen (APAP) overdose is the leading cause of ALF in Europe and North America and responsible for more cases of ALF than all other aetiologies combined (*Bernal and Wendon, 2013*; *Jaeschke, 2015*). It is estimated that each week, more than 50 million Americans use products containing APAP and approximately 30,000 patients are admitted to intensive care units every year due to APAP-induced liver injury (AILI) (*Bernal and Wendon, 2013*; *Blieden et al., 2014*). Although *N*-acetylcysteine (NAC) can prevent liver injury if given in time, there are still 30% of patients who do not respond to NAC (*Fisher and Curry, 2019*). Thus, identification of novel therapeutic targets and strategies is imperative.

APAP is metabolized predominantly by cytochrome P450 2E1 (CYP2E1) to a reactive toxic metabolite, *N*-acetyl-*p*-benzoquinone imine (NAPQI). NAPQI causes mitochondrial dysfunction, lipid peroxidation, and eventually cell death (*Hinson et al., 2004*). The initial direct toxicity of APAP triggers the cascades of coagulation and inflammation, contributing to the progression and exacerbation of AILI (*Hinson et al., 2004*). In patients with APAP overdose, the clinical observations of thrombocytopenia, reduced plasma fibrinogen levels, elevated thrombin-antithrombin, and increased levels of pro-coagulation microparticles strongly suggest concurrent coagulopathy (*Stravitz et al., 2013*; *Stravitz et al., 2016*). Similarly, APAP challenge in mice causes a rapid activation of the coagulation cascade and significant deposition of fibrin(ogen) in the liver (*Groeneveld et al., 2020*; *Sullivan et al., 2012*; *Sullivan et al., 2013*). With regard to the role of platelets in AILI, it is reported that in mice APAP-induced thrombocytopenia correlates with the accumulation of platelets in the liver and that platelet depletion significantly attenuates AILI (*Miyakawa et al., 2015*). Two recent studies also demonstrate that persistent platelet accumulation in the liver delays tissue repair after AILI in mice (*Chauhan et al., 2020*; *Groeneveld et al., 2020*). These findings strongly indicate that hepatic platelet accumulation is a key mechanism contributing to AILI. However, little is known about the underlying molecular mechanism of APAP-induced hepatic platelet accumulation and whether targeting this process could attenuate AILI.

Chitinase 3-like-1 (Chi3l1) (YKL-40 in humans) is a chitinase-like soluble protein without chitinase activities (*Lee et al., 2011*). It is produced by multiple cell types, including macrophages, neutrophils, fibroblasts, synovial cells, endothelial cells, and tumor cells (*Hakala et al., 1993*; *Kawada et al., 2007*). Chi3l1 has been implicated in multiple biological processes including apoptosis, inflammation, oxidative stress, infection, and tumor metastasis (*Lee et al., 2009*). Elevated serum levels of Chi3l1 have been observed in various liver diseases, such as hepatic fibrosis, non-alcoholic fatty liver, alcoholic liver disease, and hepatocellular carcinoma (*Kumagai et al., 2016*; *Lee et al., 2011*; *Nøjgaard et al., 2003*; *Wang et al., 2020*). However, the biological function of Chi3l1 in liver disease is not clear. Our previous study revealed an important role of Chi3l1 in promoting intrahepatic coagulation in concanavalin A-induced hepatitis (*Shan et al., 2018*). Given the importance of intrahepatic coagulation in the mechanism of AILI, we wondered whether Chi3l1 is involved in platelets accumulation during AILI.

In the current study, we observed elevated levels of Chi3l1 in patients with APAP-induced ALF and in mice challenged with APAP overdose. Our data demonstrated a central role of Chi3l1 in APAP-induced hepatic platelet recruitment through CD44. Importantly, we found that targeting Chi3l1 by monoclonal antibodies could effectively inhibit platelet accumulation in the liver and markedly attenuate AILI.

**eLife digest :** Acetaminophen, also called paracetamol outside the United States, is a commonly used painkiller, with over 50 million people in the United States taking the drug weekly. While paracetamol is safe at standard doses, overdose can cause acute liver failure, which leads to 30,000 patients being admitted to emergency care in the United States each year. There is only one approved antidote to overdoses, which becomes significantly less effective if its application is delayed by more than a few hours. This has incentivized research into identify new drug targets that could lead to additional treatment options.

Acetaminophen overdose triggers blood clotting and inflammation, contributing to liver injury. It also causes a decrease in cells called platelets circulating in the blood, which has been observed in both mice and humans. In mice, this occurs because platelets accumulate in the liver. Removing these excess cells appears to reduce the severity of the damage caused by acetaminophen, but it remains unclear how the drug triggers their accumulation in the liver. In 2018, researchers showed that a protein called Chi3l1 plays an important role in another form of liver damage. Shan et al. – including many of the researchers involved in the 2018 study – have examined whether the protein also contributes to acetaminophen damage in the liver.

Shan et al. showed that mice lacking the gene that codes for Chi3l1 developed less severe liver injury and had fewer platelets in the liver following acetaminophen overdose. They also found that human patients with acute liver failure due to acetaminophen had high levels of Chi3l1 and significant accumulation of platelets in the liver. To test whether damage could be prevented, Shan et al. used antibodies to neutralize Chi3l1 in mice after giving them an acetaminophen overdose. This reduced platelet accumulation in the liver and the associated damage.

These findings suggest that targeting Chi3l1 may be an effective strategy to prevent liver damage caused by acetaminophen overdose. Further research could help develop new treatments for acetaminophen-induced liver injury and perhaps other liver conditions.

# Materials and methods

**Key resources table**

| Reagent type (species) or resource | Designation | Source or reference | Identifiers | Additional information |
|---|---|---|---|---|
| Genetic reagent (*Mus musculus*) | C57BL/6J | Jackson Laboratory, PMID:14759567 | Stock #:000664 MGI Cat# 3849035, RRID:MGI:3849035 | |
| Genetic reagent (*Mus musculus*) | $Cd44^{-/-}$ mice (also called $Cd44^{tm1Hbg}$/$Cd44^{tm1Hbg}$) | Jackson Laboratory | Stock #:005878 MGI Cat# 4941902, RRID:MGI:4941902 | |
| Genetic reagent (*Mus musculus*) | $Chil1^{-/-}$ mice | PMID:19414556 | MGI #:3846223 RRID:MGI:3846223 | Dr Jack A. Elias (Brown University) |
| Chemical compound, drug | Acetaminophen | Sigma-Aldrich | A7085 | 210 mg/kg for male mice, 325 mg/kg for female mice |
| Peptide, recombinant protein | Recombinant mouse Chi3l1 | Sino Biological | 50929-M08H | 500 ng/mouse in 100 µl PBS |
| Commercial assay or kit | ALT diagnostic assay kit | Teco Diagnostics, | A526-120 | |
| Antibody | Syrian hamster polyclonal IgG Ctrl IgG for anti-podoplanin antibody | Bioxcell InvivoMab | BE0087, RRID:AB_1107782 | 100 µg/mouse |

*Continued on next page*

*Continued*

| Reagent type (species) or resource | Designation | Source or reference | Identifiers | Additional information |
|---|---|---|---|---|
| Antibody | Syrian hamster monoclonal anti-mouse podoplanin antibody | Bioxcell InvivoMab | BE0236, RRID:AB_2687718 | 100 µg/mouse |
| Antibody | Rat monoclonal Ctrl IgG for anti-mouse CD41 antibody | BD Biosciences | 553922, Clone R334, RRID:AB_479672 | 2 mg/kg |
| Antibody | Rat monoclonal anti-mouse CD41 antibody | BD Biosciences | 553847, Clone MWReg 30, RRID:AB_395084 | 2 mg/kg, 1:200 for IF |
| Peptide, recombinant protein | Recombinant human Chi3l1 | Sino Biological | 11227-H08H | 1 µg/mouse in 100 µl |
| Antibody | Rabbit polyclonal anti-human CD41 | Proteintech | 24552–1-AP, RRID:AB_2879604 | 1:200 for IHC |
| Antibody | Mouse monoclonal anti-human CD68 | Thermo Fisher | MA5-13324, RRID:AB_10987212 | 1:100 for IHC |
| Antibody | Rabbit polyclonal anti-human Chi3l1 | Proteintech | 12036–1-AP, RRID:AB_2877819 | 1:100 for IHC |
| Antibody | Rat monoclonal anti-mouse F4/80, Alexa 647 conjugated | Biolegend | 123122, RRID:AB_893480 | 1:100 for IF |
| Antibody | Rat monoclonal anti-mouse CD44 | abcam | ab112178, clone KM81, RRID:AB_10864553 | 1:200 for IF |
| Antibody | Golden Syrian Hamster monoclonal anti-mouse podoplanin | Novus, biological | NB600-1015, RRID:AB_2161937 | 1:100 for IF |
| Antibody | Rabbit polyclonal anti-mouse Clec-2 | Biorbyt | orb312182, RRID:AB_2891123 | 1:100 for IF |
| Antibody | Donkey anti-rat polyclonal immunoglobulin, Alexa 488-conjugated | Invitrogen | A-21208, RRID:AB_141709 | 1:1000 for IF |
| Antibody | Goat anti-rabbit polyclonal immunoglobulin, Alexa 488-conjugated | Invitrogen | A-11034, RRID:AB_141709 | 1:1000 for IF |
| Antibody | Goat anti-rabbit polyclonal immunoglobulin, Alexa 594-conjugated | Invitrogen | A-11012, RRID:AB_141359 | 1:1000 for IF |
| Antibody | Goat anti-hamster polyclonal immunoglobulin, Alexa 594-conjugated | Invitrogen | A-21113, RRID:AB_2535762 | 1:1000 for IF |
| Antibody | Hoechst | Invitrogen | H3570, RRID:AB_10626776 | 1:10000 for IF |
| Peptide, recombinant protein | TRITC-labeled Albumin | Sigma-Aldrich | A2289-10MG RRID:AB_2891111 | 10 µl/mouse for intravital microscopy |

*Continued*

| Reagent type (species) or resource | Designation | Source or reference | Identifiers | Additional information |
|---|---|---|---|---|
| Antibody | Rat monoclonal anti-mouse anti-F4/80 antibody, BV421-labeled | Biolegend | 123132, RRID:AB_11203717 | 15 µl/mouse for intravital microscopy |
| Antibody | Rat monoclonal anti-mouse CD41 antibody, DyLight 649-labeled | emfret ANALYTICS | X649, RRID:AB_2861336 | 30 µl/mouse for intravital microscopy |

## Animal experiments and procedures

C57BL/6J (RRID:MGI:3849035) and *Cd44*[-/-] mice (RRID:MGI:4941902) were purchased from the Jackson Laboratory. *Chil1*[-/-] mice were provided by Dr Jack Elias (Brown University, Providence, RI, RRID: MGI:3846223). All mouse colonies were maintained at the animal core facility of University of Texas Health Science Center (UTHealth). C57BL/6J, not C57BL/6N, was used as wild-type (WT) control because both *Chil1*[-/-] and *Cd44*[-/-] mice are on the C57BL/6J background, determined by polymerase chain reaction (PCR) (data not shown). Animal studies described have been approved by the UTHealth Institutional Animal Care and Use Committee (IACUC). For APAP treatment, mice (8–12 weeks of age) were fasted overnight (5:00 p.m. to 9:00 a.m.) before intraperitoneally (*i.p.*) injected with APAP (Sigma, A7085) at a dose of 210 mg/kg for male mice and 325 mg/kg for female mice, as female mice are less susceptible to AILI (*Munoz and Fearon, 1984*). Male mice have been the choice in the vast majority of the studies of AILI reported in the literature (*Ju et al., 2002*; *Sullivan et al., 2012*). Therefore, we used male mice in the majority of the experiments presented. In some experiments, APAP-treated mice were immediately injected *i.p.* with either PBS (100 µl) or recombinant mouse Chi3l1 (rmChi3l1, 500 ng/mouse in 100 µl, Sino Biological 50929-M08H). Livers were harvested at time points indicated in the figure legends and immunofluorescence (IF) staining was performed using frozen sections to detect MΦs and platelets using anti-F4/80 and anti-CD41 antibodies, respectively. Liver paraffin sections and sera were harvested at time points indicated in the figure legends. H&E staining and ALT measurement to examine liver injury were performed using a diagnostic assay kit (Teco Diagnostics, Anaheim, CA).

## Blocking endogenous podoplanin

Mice were intravenously (*i.v.*) injected with Ctrl IgG (Bioxcell InvivoMab, BE0087, 100 µg/mouse, RRID:AB_1107782) or anti-podoplanin antibody (Bioxcell InvivoMab, BE0236, 100 µg/mouse, RRID: AB_2687718) in *Chil1*[-/-] reconstituted with rmChi3l1 at 16 hr prior to APAP treatment.

## Platelet depletion

WT mice were *i.v.* injected with Ctrl IgG (BD Pharmingen, 553922, 2 mg/kg, RRID:AB_479672) or CD41 antibody (BD Pharmingen, 553847, 2 mg/kg, RRID:AB_395084) to deplete platelets at 12 hr prior to APAP treatment.

## Kupffer cells depletion

WT mice were *i.v.* injected with empty liposomes (PBS, 100 µl/mouse) or clodronate (CLDN)-containing liposomes (100 µl/mouse) to deplete Kupffer cells (KCs) at either 9 or 40 hr prior to APAP treatment. CLDN-containing liposomes were generated as previously described (*Ju et al., 2002*).

## Evaluation of the effects of anti-Chi3l1 monoclonal antibodies

To examine the therapeutic potential of anti-mouse Chi3l1 monoclonal antibodies (mAbs), WT mice were injected (*i.p.*) with either Ctrl IgG or anti-mouse Chi3l1 antibody clones 3 hr after APAP administration. To examine the therapeutic potential of anti-human Chi3l1 mAbs, *Chil1*[-/-] mice treated with APAP were immediately injected (*i.p.*) with either PBS (100 µl) or recombinant human Chi3l1 (rhChi3l1, 1 µg/mouse in 100 µl, Sino Biological 11227-H08H). After 3 hr, these mice were divided into two groups injected (*i.p.*) with either Ctrl IgG or anti-human Chi3l1 mAbs.

## Bio-layer interferometry

The binding affinity between Fc-CD44 and His-Chi3l1 was measured using the Octet system eight-channel Red96 (Menlo Park). Protein A biosensors and kinetics buffer were purchased from Pall Life Sciences (Menlo Park). Fc-CD44 protein was immobilized onto protein A biosensors and incubated with varying concentrations of recombinant His-Chi3l1 in solution (1000–1.4 nM). Binding kinetic constants were determined using 1:1 fitting model with ForteBio's data analysis software 7.0, and the KD was calculated using the ratio Kdis/Kon (the highest four concentrations were used to calculate the KD).

## IHC and IF

H&E staining and immunohistochemistry (IHC) were performed on paraffin sections using the following antibodies: anti-human CD41 (Proteintech, 24552–1-AP, 1:200, RRID:AB_2879604), anti-human CD68 (Thermo Fisher, MA5-13324, 1:100, RRID:AB_10987212), anti-human Chi3l1 (Proteintech, 12036–1-AP, 1:100, RRID:AB_2877819). IF staining was performed on frozen sections using the following antibodies: anti-mouse CD41 (BD Bioscience, Clone MWReg 30, RRID:AB_395084), mouse F4/80 (Biolegend, 123122, 1:100, RRID:AB_893480), anti-CD44 (abcam, clone KM81, ab112178, 1:200, RRID:AB_10864553), anti-Chi3l1 (Proteintech, 12036–1-AP, 1:100, RRID:AB_2877819), anti-podoplanin (Novus, NB600-1015, 1:100, RRID:AB_2161937), and anti-Clec-2 (C-type lectin-like receptor 2) (Biorbyt, orb312182, 1:100, RRID:AB_2891123). Alexa 488-conjugated donkey anti-rat immunoglobulin (Invitrogen, A-21208, 1:1000, RRID:AB_141709) was used as a secondary antibody for CD41 and CD44 detection. Alexa 488-conjugated goat anti-rabbit immunoglobulin (Invitrogen, A-11034, 1:1000, RRID:AB_141709) was used as a secondary antibody for Clec-2 detection. Alexa 594-conjugated goat anti-rabbit immunoglobulin (Invitrogen, A-11012, 1:1000, RRID:AB_141359) was used as a secondary antibody for Chi3l1 detection. Alexa 594-conjugated goat anti-hamster immunoglobulin (Invitrogen, A-21113, 1:1000, RRID:AB_2535762) was used as a secondary antibody for podoplanin detection. Nuclei were detected by Hoechst (Invitrogen, H3570, 1:10,000, RRID:AB_10626776).

## Intravital confocal microscopy

Mice were prepared for intravital microscopy as previously described (*Da et al., 2018*). Briefly, mice were anesthetized using pentobarbital and underwent tracheostomy (to facilitate breathing) and internal jugular catheterization (for antibody administration) followed by liver exteriorization as described by *Marques et al., 2015* with modifications. Mice were placed supine on a custom-made stage with the liver overlying a glass coverslip wetted with warmed saline and surrounded with wet saline-soaked gauze. Mice were kept euthermic at 37°C using radiant warmers and monitored with a rectal thermometer. Anesthesia was maintained using an isoflurane delivery device (RoVent with SomnoSuite; Kent Scientific) with 1–3% isoflurane delivered. Mice were *i.v.* injected with an antibody mixture in sterile 0.9% sodium chloride containing TRITC/bovine serum albumin (Sigma; to label the vasculature; 500 µg/mouse, RRID:AB_2891111), BV421-anti-F4/80 antibody (to label Kupffer; 0.75 µg/mouse, RRID:AB_11203717), and DyLight 649/anti-GPIbβ antibody (emfret analytics; to label platelets; 3 µg/mouse, RRID:AB_2861336) for visualization. Mice were imaged on an Olympus FV3000RS laser scanning confocal inverted microscope system at 30 fps using a 60×/NA1.30 silicone oil objective with 1× and 3× optical zoom using the resonance scanner. This allows for simultaneous excitation and detection of up to four wavelengths. All animals were euthanized under a surgical plane of anesthesia at the end of the experiments.

## Image analysis of intravital microscopy experiments

The images were then analyzed by a blinded investigator to assess platelet area. Eleven to fifteen 1 min fields of view (1× optical zoom) were analyzed per mouse using FIJI/ImageJ software. Background noise was removed using a Gaussian filter (one pixel) for all channels prior to analysis. Vascular area was measured in each field using the region of interest selection brush in the TRITC (albumin) channel. The platelet area within the vascular ROI was then determined using threshold of the DyLight 649 (platelet) channel.

## Generation of Chi3l1 mAbs

Rabbit mAbs were generated using previously reported methods (*Deng et al., 2018*). Briefly, two New Zealand white rabbits were immunized subcutaneously with 0.5 mg recombinantly expressed human Chi3l1 protein (Sino Biological, Cat#11227-H08H). After the initial immunization, animals were given boosters three times in a 3-week interval. Serum titers were evaluated by indirect enzyme-linked immunosorbent assay (ELISA) and rabbit peripheral blood mononuclear cells (PBMCs) were isolated after the final immunization. A large panel of single memory B cells were enriched from the PBMCs and cultured for 2 weeks, and the supernatants were analyzed by ELISA. To isolate mouse Chi3l1 antibody, the rabbits were boosted twice more with mouse Chi3l1 before the memory B cell culture. The variable region genes of the antibodies from these positive single B cells were recovered by reverse transcription PCR (RT-PCR) and cloned into the mammalian cell expression vector as described previously (*Deng et al., 2018*). Both the heavy and light chain constructs were co-transfected into Expi293 cell lines using transfection reagent PEI (Sigma). After 7 days of expression, supernatants were harvested and antibodies were purified by affinity chromatography using protein A resin as reported before (*Deng et al., 2018*).

## Statistics

Data were presented as mean ± SEM unless otherwise stated. Statistical analyses were carried out using GraphPad Prism (GraphPad Software). Comparisons between two groups were carried out using unpaired Student's t-test. Comparisons among multiple groups ($n \geq 3$) were carried out using one-way ANOVA. p-Values are as labeled and less than 0.05 was considered significant. Platelets counts/mm$^2$ was analyzed using ImageJ software.

## Study approval

Serum samples from patients diagnosed with APAP-induced liver failure on day 1 of admission were obtained from the biobank of the Acute Liver Failure Study Group (ALFSG) at UT Southwestern Medical Center, Dallas, TX, USA. The study was designed and carried out in accordance with the principles of ALFSG and approved by the Ethics Committee of ALFSG (HSC-MC-19–0084). Formalin-fixed, paraffin-embedded human liver biopsies from patients diagnosed with APAP-induced liver failure were obtained from the National Institutes of Health-funded Liver Tissue Cell Distribution System at the University of Minnesota, which was funded by NIH contract # HHSN276201200017C.

See Materials and methods for details for other methods.

## Blocking endogenous CD44

Mice were *i.p.* injected with Ctrl IgG (BD Pharmingen, 559478, 50 µg/mouse) or anti-CD44 antibody (BD Pharmingen, 553131, 50 µg/mouse) in *Chil1*$^{-/-}$ reconstituted with rmChi3l1 at 30 min prior to APAP treatment.

## Preparation of liver cells and in vitro cell culture

Hepatic NPCs and hepatocytes were isolated as previously described (*Shan et al., 2018*). In brief, mice were anesthetized and liver tissues were perfused with EGTA solution, followed by a 0.04% collagenase digestion buffer. Liver hepatocytes and NPCs were isolated by gradient centrifugation using 35% percoll (Sigma). To further purify LSEC and MΦs, LSEC and MΦs fractions were stained with phycoerythrin (PE)-conjugated anti-CD146 (for LSEC, Invitrogen, 12-1469-42), and anti-F4/80 (for MΦs, Invitrogen, 12-4801-82) antibodies and positively selected using EasySep Mouse PE Positive Selection Kit (Stemcell Technologies) following manufacturer's instructions. Each subset will yield a purity around 90%.

## Co-culture of MΦs and platelets

Isolated MΦs were cultured in DMEM with 10% fetal bovine serum and pre-treated with podoplanin antibody (Bioxcell InvivoMab, BE0236, 2 µg/ml) for 30 min and then co-culture with washed platelets for 30 min. Unbound platelets were washed out and podoplanin and Clec-2 on MΦs were stained.

## Isolation of platelets

Mouse whole blood was collected with anti-coagulant ACD solution from Inferior vena cava. Platelets were further isolated by additional washes with Tyrode's buffer. Isolated washed platelets were subjected to functional assay after incubation with $PGI_2$ (Sigma, P6188) for 30 min.

## Flow cytometry

Isolated liver NPCs were incubated with1 μl of anti-mouse FcγRII/III (Becton Dickinson, Franklin Lakes, NJ) to minimize non-specific antibody binding. The cells were then stained with anti-mouse CD45-V655 (eBioscience, 15520837), F4/80-APC/Cy7 (Biolegend, 123118), Ly6C-APC (BD Pharmingen, 560595), Ly6G-V450 (BD Pharmingen, 560603), CD146-PerCP-Cy5.5 (BD Pharmingen, 562134), CD44-PE (BD Pharmingen, 553134), anti-His-FITC (abcam, ab1206). In some experiments, cells were incubated with 2 μg rmChi3l1 for 2 hr before antibody staining. The cells were analyzed on a Cyto-FLEX LX Flow Cytometer (Beckman Coulter, Indianapolis, IN) using FlowJo software (Tree Star, Ashland, OR). For flow cytometric analysis, $CD45^+$ cells were gated to exclude endothelial cells, hepatic stellate cells, and residue hepatocytes. Within $CD45^+$ cells, $CD44^+$ cells that bind to Chi3l1 were back-gated to determine the cells types.

## Extraction of liver proteins, immunoprecipitation, and mass spectrometry

Snap-frozen liver tissues were pulverized to extract liver proteins in STE buffer. Protein concentration was measured by BCA kit (Thermo Scientific, 23225) following the manufacturer's instructions.

## Immunoprecipitation of NPCs lysates

Proteins were extracted from NPCs lysates and incubated with 5 μg rmChi3l1, followed by immunoprecipitation with 2 μg rabbit IgG (negative control, Peprotech, 500-p00) or 2 μg anti-His tag antibody (Abnova, MAB12807). Dynabeads Protein G (Invitrogen, 1003D) were used to pull down antibodies-binding proteins. Immunoprecipitated proteins were subject to mass spectrometry analyses by the Proteomics Core Facility at UTHealth.

## Immunoprecipitation of liver homogenates

$Cd44^{-/-}$ and WT mice were treated with APAP for 2 hr; 10 mg liver proteins were extracted from treated mice and incubated with 5 μg rmChi3l1, followed by immunoprecipitation with 2 μg anti-CD44 antibody (BD Pharmingen, 553131). Dynabeads Protein G (Invitrogen, 1003D) were used to pull down antibody-binding proteins. Input and immunoprecipitated proteins were subject to Western blot analyses.

## In vitro immunoprecipitation assays

Two microgram rhChi3l1 (Sino Biological, His Tag, 11227-H08H) or 2 μg GST protein (His Tag) as control were incubated with 2 μg human CD44 (Sino Biological, Fc Tag, 12211-H02H) and immunoprecipitated with 2 μg anti-His antibody (Abnova, MAB12807). Input and immunoprecipitated proteins were subject to Western blot analyses.

## Western blotting

Samples were prepared with loading buffer and boiled before loading onto SDS-PAGE gels. Nitrocellulose membranes (Bio-Rad) were used to transfer proteins. Primary antibodies used to detect specific proteins: anti-Chi3l1 (Proteintech, 12036–1-AP, 1:1000), anti-CD44 (abcam, ab25340, 1:500), anti-β-actin (Cell Signaling, 4970, 1:1000), anti-His (Abnova, MAB12807, 1:1000), anti-cyp2e1 (LifeSpan BioSciences, LS-C6332, 1:500), anti-APAP adducts (*Ju et al., 2002*) (provided by Dr Lance R. Pohl, NIH, 1:500). Secondary antibodies include goat anti-Rabbit IgG (Jackson ImmunoResearch, 111-035-144, 1:1000), goat anti-Rat (Jackson ImmunoResearch, 112-035-003, 1:1000).

## Quantitative real-time RT-PCR

Total RNA was isolated from $1 \times 10^6$ cells using RNeasy Mini Kit (Qiagen, Valencia, CA). After the removal of genomic DNA, RNA was reversely transcribed into cDNA using Moloney murine leukemia virus RT (Invitrogen, Carlsbad, CA) with oligo (dT) primers (Invitrogen). Quantitative PCR was

performed using SYBR green master mix (Applied Biosystem) in triplicates on a Real-Time PCR 7500 SDS system and software following manufacturer's instruction (Life Technologies, Grand Island, NY). RNA content was normalized based on amplification of 18S ribosomal RNA (rRNA) (18S). Change folds = normalized data of experimental sample/normalized data of control. The specific primer pairs used for PCR are listed in *Table 1*.

## Results

### Chi3l1 is upregulated and plays a critical role in AILI

Although elevated serum levels of Chi3l1 have been observed in chronic liver diseases (*Kumagai et al., 2016*; *Lee et al., 2011*; *Nøjgaard et al., 2003*; *Wang et al., 2020*), modulations of Chi3l1 levels during acute liver injury have not been reported. Our data demonstrated, for the first time, that compared with healthy individuals, patients with AILI displayed higher levels of Chi3l1 in the liver and serum (*Figure 1A,B*). Similarly, in mice treated with APAP, hepatic mRNA and serum protein levels of Chi3l1 were upregulated (*Figure 1C,D*). To determine the role of Chi3l1 in AILI, we treated WT mice and Chi3l1-knockout (*Chil1*[-/-]) mice with APAP. Compared with WT mice, serum ALT levels and the extent of liver necrosis were dramatically lower in *Chil1*[-/-] mice (*Figure 1E, F*). Moreover, administration of rmChi3l1 protein to *Chil1*[-/-] mice enhanced liver injury to a similar degree observed in APAP-treated WT mice (*Figure 1E,F*). These data strongly suggest that Chi3l1 contributes to AILI.

### Chi3l1 contributes to AILI by promoting hepatic platelet recruitment

Thrombocytopenia is often observed in patients with APAP overdose (*Harrison et al., 1990*; *Stravitz et al., 2013*; *Stravitz et al., 2016*). We hypothesized that this phenomenon may be attributed to the recruitment of platelets into the liver. We performed IHC staining of liver biopsies from patients with APAP-induced liver failure and found markedly increased numbers of platelets compared with normal liver tissues (*Figure 2A*). Similarly, in mice treated with APAP, a marked increase of platelets in the liver was observed by intravital microscopy (*Figure 2B*). It is reported that depletion of platelets prior to APAP treatment can prevent liver injury in mice (*Miyakawa et al., 2015*). Our data demonstrated that even after APAP treatment, depletion of platelets could still attenuate AILI (*Figure 2C,D*; *Figure 2—figure supplement 1*). These data indicate a critical contribution of

**Table 1.** Real-time PCR primers used.

| Gene | Forward (F)/reverse (R) primer | Primer sequences |
| --- | --- | --- |
| Pdpn | F | ACCGTGCCAGTGTTGTTCTG |
| | R | AGCACCTGTGGTTGTTATTTTGT |
| Psgl-1 | F | GAAAGGGCTGATTGTGACCCC |
| | R | AGTAGTTCCGCACTGGGTACA |
| Cd40 | F | TGTCATCTGTGAAAAGGTGGTC |
| | R | ACTGGAGCAGCGGTGTTATG |
| Mcam | F | GTGGCGTTGACATCGTTGG |
| | R | CTATGTACTTCGTATGCAGGTCG |
| Icam-1 | F | GTGATGCTCAGGTATCCATCCA |
| | R | CACAGTTCTCAAAGCACAGCG |
| Fcr | F | AGGGCCTCCATCTGGACTG |
| | R | GTGGTTCTGGTAATCATGCTCTG |
| Lfa1 | F | CCAGACTTTTGCTACTGGGAC |
| | R | GCTTGTTCGGCAGTGATAGAG |
| Vwf | F | CTCTTTGGGGACGACTTCATC |
| | R | TCCCGAGAATGGAGAAGGAAC |

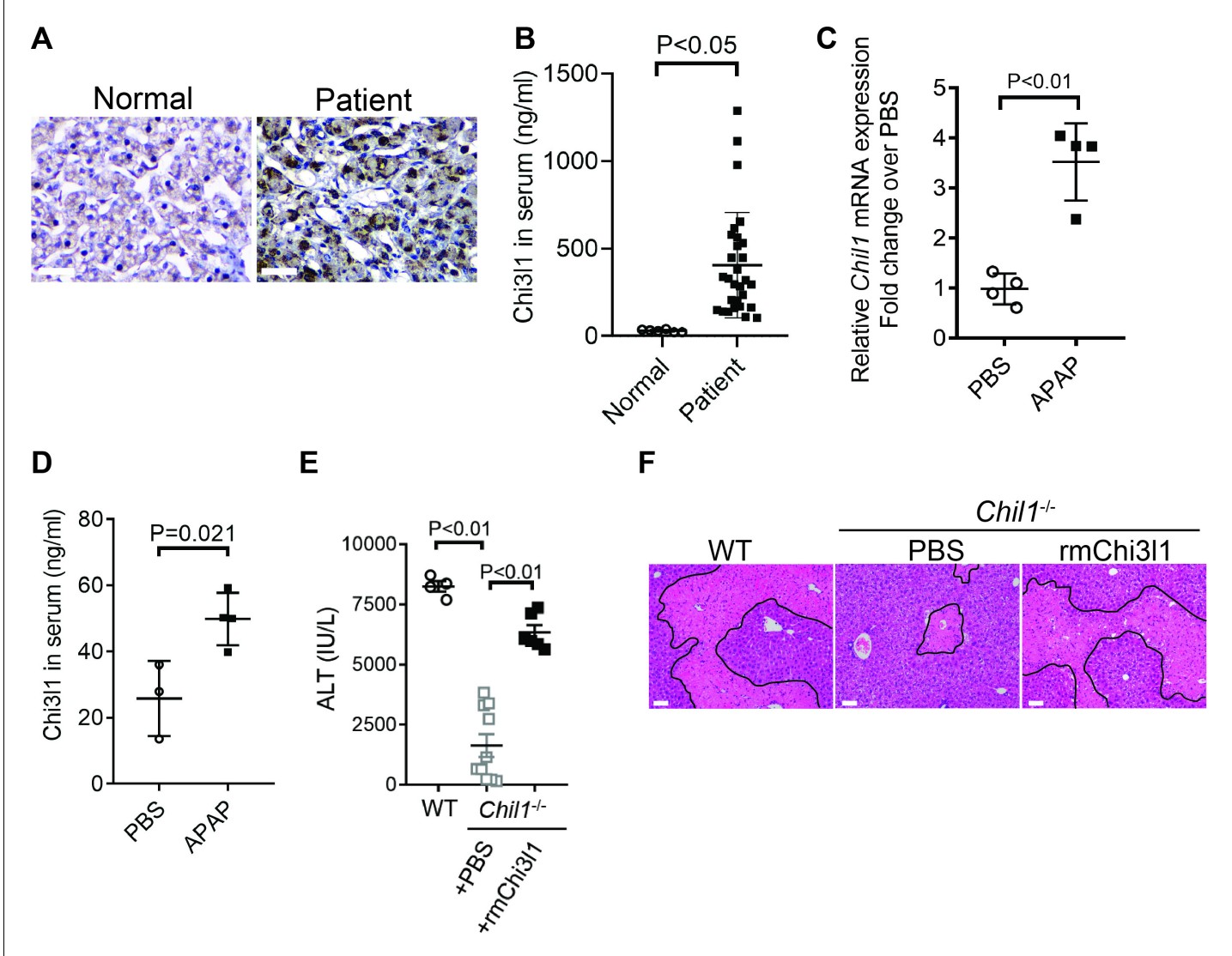

**Figure 1.** Chitinase 3-like-1 (Chi3l1) is upregulated and plays a critical role in acetaminophen-induced liver injury (AILI). (**A**) Immunohistochemical (IHC) staining for Chi3l1 in normal liver biopsies (Normal) and those from patients with AILI (Patient). Images shown are representative of 10 samples/group. Scale bar, 250 μm. (**B**) Enzyme-linked immunosorbent assay (ELISA) analysis of Chi3l1 in serum of healthy individuals (Normal, n = 6) and patients with AILI (Patient, n = 29). Data were presented as median+interquartile range. (**C, D**) Male C57B/6 mice treated with PBS or acetaminophen (APAP). (**C**) *Chil1* mRNA in liver homogenates and (**D**) Chi3l1 protein levels in serum were measured by quantitative reverse transcription polymerase chain reaction (qRT-PCR) and ELISA at 3 and 24 hr, respectively (n = 4 mice/group). (**E, F**) Male C57B/6 (wild-typr [WT]) and *Chil1*−/− mice were treated with APAP. Additionally, *Chil1*−/− mice were divided into two groups treated with either PBS or recombinant mouse Chi3l1 (rmChi3l1) simultaneously with APAP (n = 4–10 mice/group). (**E**) Serum levels of ALT and (**F**) liver histology with necrotic areas outlined were evaluated 24 hr after APAP treatment. Scale bar, 250 μm. Mann-Whitney test was performed in **B**. Two-tailed, unpaired Student's t-test was performed in **C, D**. One-way ANOVA were performed in **E**.

platelets to AILI. Given the role of Chi3l1 in promoting intrahepatic coagulation in concanavalin A-induced hepatitis (*Shan et al., 2018*), we hypothesized that Chi3l1 might be involved in platelet recruitment to the liver during AILI. To examine this hypothesis, we detected platelets in the liver by IHC using anti-CD41 antibody. Comparing with WT mice, we observed much fewer platelets in the liver after APAP treatment (*Figure 2E*). Moreover, administration of rmChi3l1 to *Chil1*−/− mice restored hepatic platelet accumulation similar to APAP-treated WT mice (*Figure 2E*). These data suggest that Chi3l1 plays a critical role in promoting hepatic platelet accumulation, thereby contributing to AILI.

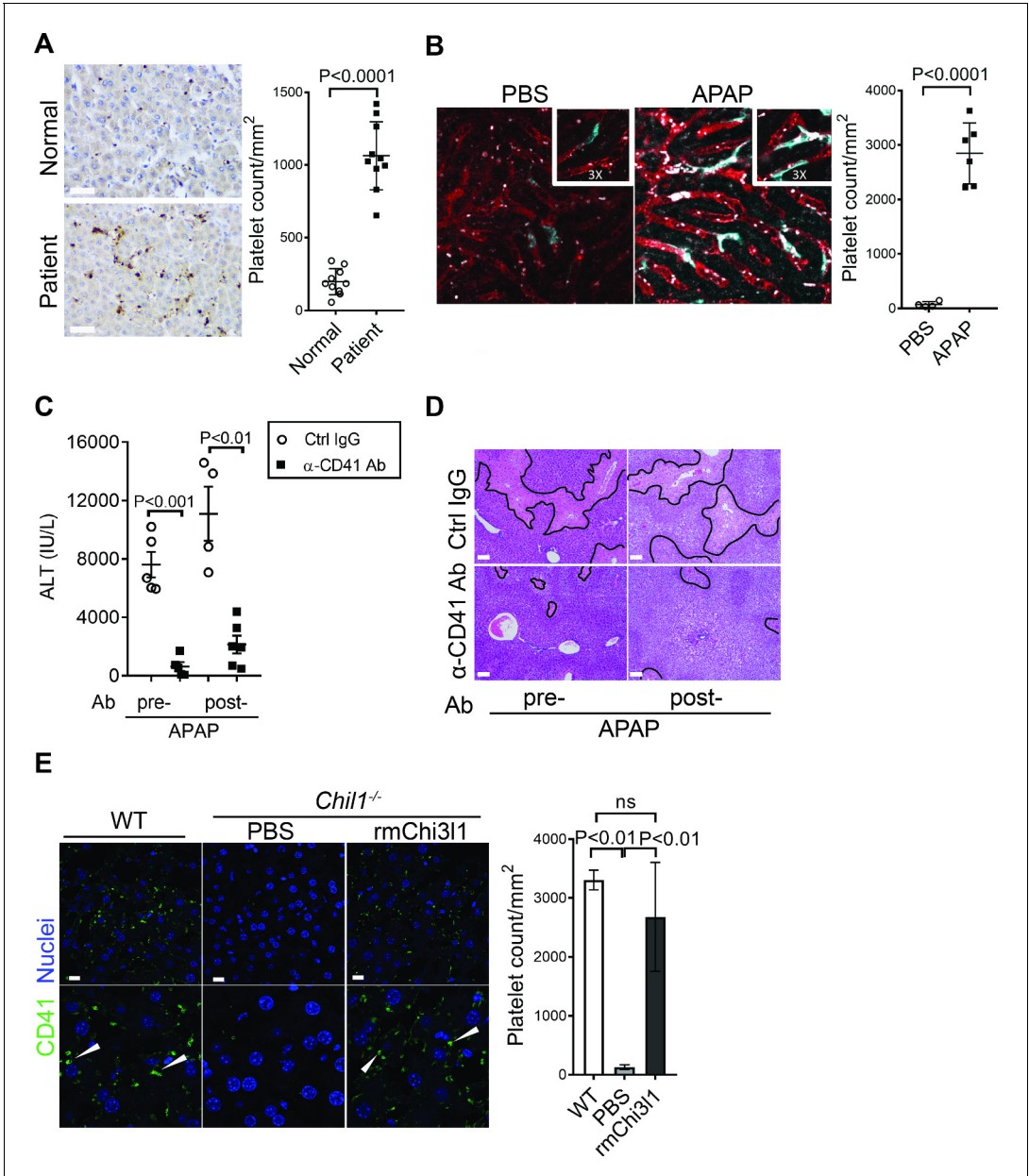

**Figure 2.** Chitinase 3-like-1 (Chi3l1) contributes to acetaminophen-induced liver injury (AILI) by promoting hepatic platelet recruitment. (**A**) Immunohistochemical (IHC) staining to detect platelets (CD41+) in healthy liver biopsies (Normal) and those from patients with AILI (Patient). Scale bar, 250 μm (n = 10/group). (**B**) Male C57B/6 mice treated with PBS or acetaminophen (APAP). Intravital microscopy analyses were performed around 3 hr post-APAP. MΦs (cyan) and platelets (white) in liver sinusoids (red) are indicated. Representative images were chosen from intravital microscopy videos: https://bcm.box.com/s/15hmtryyrdl302mihrsm034ure87x4ea (Supplementary video 1, PBS treatment) and https://bcm.box.com/s/tuljfmstvv4lvoksx16fkxkpirkekynz (Supplementary video 2; n = 6–7 mice/group, 4–15 videos/mouse). (**C–E**) Male C57B/6 (wild-type [WT]) mice were treated with control IgG (Ctrl IgG) or an anti-CD41 antibody (α-CD41 Ab) either 3 hr before or 3 hr after APAP administration. (**C**) Serum levels of ALT and (**D**) liver histology with necrotic areas outlined were evaluated 24 hr after APAP treatment (n = 5 mice/group in **C**, **D**). Scale bar, 250 μm. (**E**) Male C57B/6 (WT) and Chil1-/- mice were treated with APAP. Additionally, Chil1-/- mice were divided into two groups treated with either PBS or recombinant mouse Chi3l1 (rmChi3l1) simultaneously with APAP. Immunofluorescence (IF) staining was performed to detect intrahepatic platelets (CD41+) 3 hr after APAP treatment (n = 3 mice/group). Scale bar, 25 μm. Two-tailed, unpaired Student's t-test was performed in **A–C**. One-way ANOVA were performed in **E**.

The online version of this article includes the following figure supplement(s) for figure 2:

**Figure supplement 1.** Depletion of platelets by anti-CD41 antibody reduces hepatic platelets recruitment.

## Chi3l1 functions through its receptor CD44

To further understand how Chi3l1 is involved in platelet recruitment, we set out to identify its receptor. We isolated non-parenchymal cells (NPCs) from WT mice at 3 hr after APAP treatment and incubated the cells with His-tagged rmChi3l1. The cell lysate was subjected to immunoprecipitation using an anti-His antibody. The 'pulled down' fraction was subjected to LC/MS analyses, and a partial list of proteins identified is shown in *Supplementary file 1*. Among the potential binding proteins, we decide to further investigate CD44, which is a cell surface receptor expressed on diverse mammalian cell types, including endothelial cells, epithelial cells, fibroblasts, keratinocytes, and leukocytes (*Ponta et al., 2003*). Immunoprecipitation experiments using liver homogenates from APAP-treated WT and *Cd44*$^{-/-}$ mice demonstrated that the anti-CD44 antibody could 'pull down' Chi3l1 from WT but not *Cd44*$^{-/-}$ liver homogenates (*Figure 3A*). Supporting this finding, interferometry measurements using rhChi3l1 revealed a direct interaction between Chi3l1 and CD44 (Kd = 251 nM, *Figure 3B*). Moreover, we incubated rhChi3l1 with human CD44 and then performed immunoprecipitation with an anti-CD44 antibody. Data shown in *Figure 3C* confirmed that Chi3l1 directly binds to CD44. Together, these results suggest that CD44 is a receptor for Chi3l1.

To investigate the role of CD44 in mediating the function of Chi3l1, we treated *Cd44*$^{-/-}$ mice with rmChi3l1 simultaneously with APAP challenge. We found that rmChi3l1 had no effect on platelet recruitment or AILI in *Cd44*$^{-/-}$ mice (*Figure 3D–F*). This is in stark contrast to restoring platelet accumulation and increasing AILI by rmChi3l1 treatment in *Chil1*$^{-/-}$ mice (*Figure 1E, F* and *2E*). However, these effects of rmChi3l1 in *Chil1*$^{-/-}$ mice were abrogated when CD44 was blocked by using an anti-CD44 antibody (*Figure 3—figure supplement 1A–C*). Together, these data demonstrate a critical role of CD44 in mediating Chi3l1-induced hepatic platelet accumulation and AILI.

CYP2E1-mediated APAP bio-activation to form NAPQI and the detoxification of NAPQI by glutathione (GSH) are important in determining the degrees of AILI (*Hinson et al., 2004*). Although unlikely, there is a possibility that the phenotypes observed in *Chil1*$^{-/-}$ and *Cd44*$^{-/-}$ mice were due to the effects of gene deletion on APAP bio-activation. To address this concern, we compared the levels of GSH, liver CYP2E1 protein expression, and NAPQI-protein adducts among WT, *Chil1*$^{-/-}$ and *Cd44*$^{-/-}$ mice (*Figure 3—figure supplement 2A–C*). However, we did not observe any difference, suggesting that Chi3l1 or CD44 deletion does not affect APAP bio-activation and its direct toxicity to hepatocytes. Moreover, although we used male mice performed all of the experiments, we observed a similar phenotype in female *Chil1*$^{-/-}$ and *Cd44*$^{-/-}$ mice as in male mice (*Figure 3—figure supplement 3*).

## Hepatic MΦs promote platelet recruitment

To further identify the cell type on which Chi3l1 binds to CD44, we incubated liver NPCs with His-tagged rmChi3l1. We found that almost all CD44$^{+}$Chi3l1$^{+}$ cells were F4/80$^{+}$ MΦs (*Figure 3—figure supplement 2D*). This finding suggested the possible involvement of hepatic MΦs in platelet recruitment. We performed IHC staining of liver biopsies from AILI patients and observed co-localization of MΦs (CD68$^{+}$) and platelets (CD41$^{+}$) (*Figure 4A*). In the livers of APAP-treated mice, adherence of platelets to MΦs was also observed by IHC (*Figure 4B*) and intravital microscopy (*Figure 2B*). Quantification of the staining confirmed that there were higher numbers of platelets adherent to MΦs than to liver sinusoidal endothelial cells (LSECs) after APAP challenge (*Figure 4B*).

To further investigate the role of hepatic MΦs in platelet recruitment during AILI, we performed MΦ-depletion experiments using liposome-encapsulated CLDN. We first followed a previously published protocol (*Campion et al., 2008*; *Fisher et al., 2013*; *Ju et al., 2002*) and injected CLDN around 40 hr prior to APAP treatment (*Figure 4C*, 'Previous Strategy'). We examined the efficiency of MΦ-depletion by flow cytometry analysis, which can distinguish resident KCs (CD11b$^{low}$F4/80$^{+}$) from infiltrating MΦs (IMs, CD11b$^{hi}$F4/80$^{+}$) (*Holt et al., 2008*). We found that compared with control mice treated with empty liposomes, there were actually more MΦs, consisted of mainly IMs, in the liver of CLDN-treated mice (*Figure 4C*). Consistent with the increase of MΦs, there were also higher numbers of platelets in the liver of CLDN-treated mice (*Figure 4C*). These findings suggest that although KCs are depleted using the 'Previous Strategy', the treatment of CLDN induces the recruitment of IMs, resulting in higher numbers of MΦs in the liver at the time of APAP treatment. As reported, this treatment strategy resulted in exacerbated AILI (*Figure 4E,F*, 'Previous Strategy'), which had led to the conclusion in published reports that KCs play a protective role against

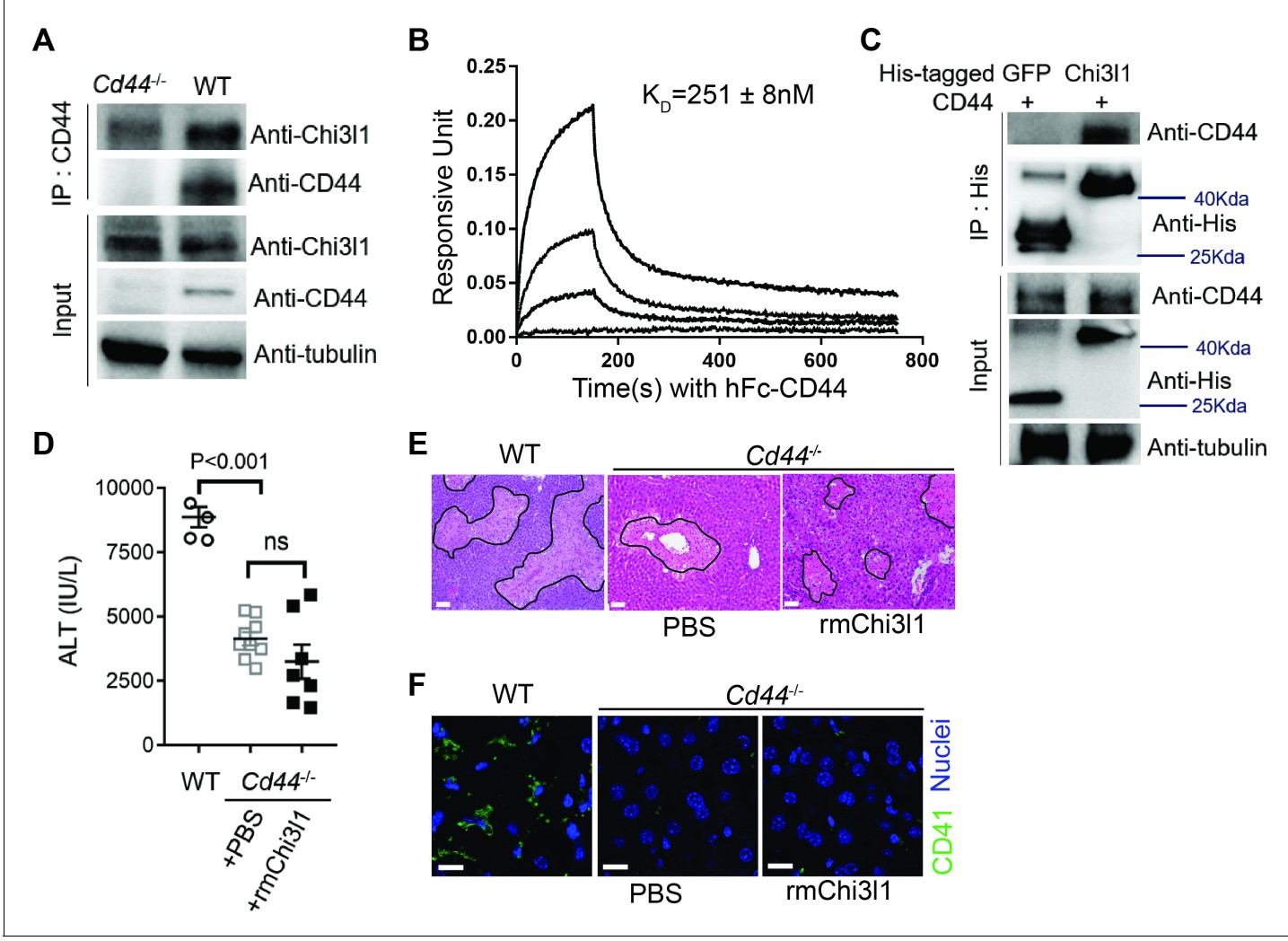

**Figure 3.** Chitinase 3-like-1 (Chi3l1) functions through its receptor CD44. (**A**) Immunoprecipitation with anti-CD44 antibody was performed using liver homogenates obtained from wild-type (WT) and *Cd44⁻/⁻* mice treated with acetaminophen (APAP) for 2 hr. Input proteins and immune-precipitated proteins were blotted with the indicated antibodies. (**B**) Interferometry measurement of the binding kinetics of human His-Chi3l1 with human Fc-CD44. (**C**) His-tagged control GFP and human Chi3l1 were incubated with recombinant human CD44. Proteins bound to Chi3l1 were immune-precipitated with an anti-His antibody. Input proteins and immune-precipitated proteins were blotted with indicated antibodies. (**D–F**) Male WT mice were treated with APAP and *Cd44⁻/⁻* mice were treated with PBS or recombinant mouse Chi3l1 (rmChi3l1) plus APAP. (**D**) Serum levels of ALT and (**E**) liver histology with necrotic areas outlined were evaluated 24 hr after APAP treatment (n = 4–9 mice/group in A, B). Scale bar, 250 µm. (**F**) Immunofluorescence (IF) staining was performed to detect intrahepatic platelets (CD41⁺) 3 hr after APAP treatment (n = 3 mice/group). Scale bar, 25 µm. One-way ANOVA were performed in D.

The online version of this article includes the following figure supplement(s) for figure 3:

**Figure supplement 1.** Chitinase 3-like-1 (Chi3l1) promotes hepatic platelet recruitment and acetaminophen-induced liver injury (AILI) through CD44 expressing on MΦs.

**Figure supplement 2.** Deletion of neither chitinase 3-like-1 (Chi3l1) nor CD44 affects acetaminophen (APAP) bio-activation.

**Figure supplement 3.** Female *Chil1⁻/⁻* and *Cd44⁻/⁻* mice develop reduced liver injury compared to female wild-type (WT) mice.

AILI (*Campion et al., 2008*; *Fisher et al., 2013*; *Ju et al., 2002*). However, alternatively the enhanced injury could be due to increased IMs and platelet accumulation.

To better investigate the role of hepatic MΦs in platelet recruitment, we set out to identify a time period in which both KCs and IMs are absent after CLDN treatment. We measured hepatic MΦs by flow cytometry at various time points after CLDN treatment and established a 'New Strategy', in which mice were injected with CLDN and after 9 hr treated with APAP. As shown in *Figure 4D*, at 6 hr after APAP challenge (15 hr after CLDN), both KCs and IMs were dramatically reduced.

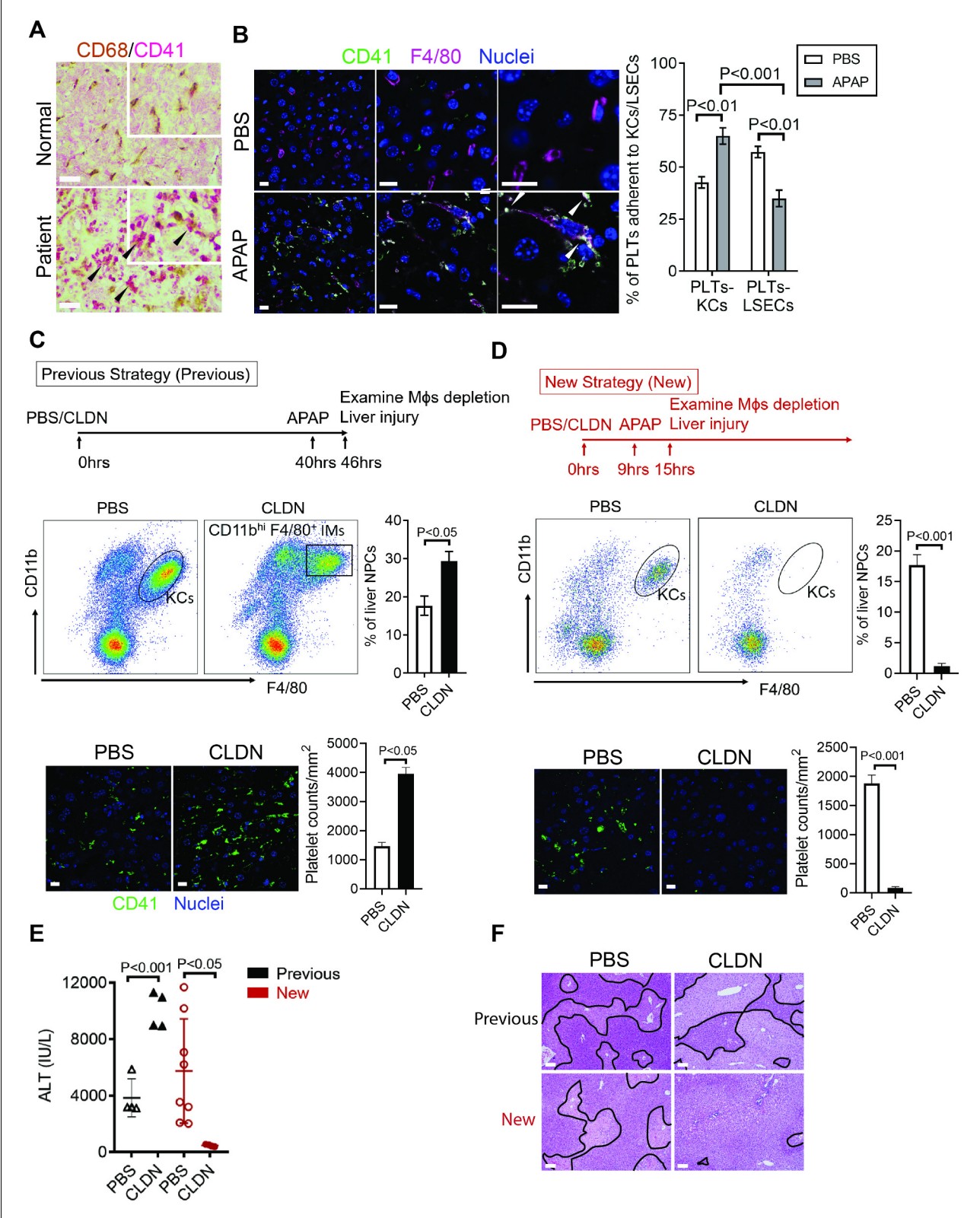

**Figure 4.** Hepatic MΦs promote platelet recruitment. (**A**) Immunohistochemical (IHC) staining for macrophages (CD68[+]) and platelets (CD41[+]) in normal liver biopsies (Normal) and those from patients with AILI (Patient) (n = 10/group). Scale bar, 25 μm. (**B**) Immunofluorescence (IF) staining for intrahepatic platelets (CD41[+]) and Kupffer cells (KCs) (F4/80[+]) in male C57B/6 mice treated with PBS or acetaminophen (APAP) for 3 hr. Scale bar, 25 μm. Arrowheads indicate platelets adherent to KCs. Quantification of platelets adherent to KCs or liver sinusoidal endothelial cells (LSECs). (**C–F**) Male

*Figure 4 continued on next page*

Figure 4 continued

C57B/6 mice were injected with either empty liposomes containing PBS (PBS) or liposomes containing clodronate (CLDN), followed by APAP treatment. (C, D) Non-parenchymal cells (NPCs) were isolated and underwent flow cytometry analysis. Indicated cells were gated on single live CD45+CD146- cells. IF staining was performed to detect intrahepatic platelets (CD41+). Scale bar, 25 µm. (E) Serum levels of ALT and (F) liver histology with necrotic areas outlined. Scale bar, 250 µm (n = 6 mice/group in B-F). Two-tailed, unpaired Student's t-test was performed in B-D, F.

Interestingly, when compared to control mice treated with empty liposomes, CLDN-treated mice developed markedly reduced liver injury with nearly no platelet accumulation in the liver (*Figure 4D–F*, 'New Strategy'). These data suggest that hepatic MΦs play a crucial role in platelet recruitment into the liver, thereby contributing to AILI.

## Chi3l1/CD44 signaling in MΦs upregulates podoplanin expression and platelet adhesion

To further understand how Chi3l1/CD44 signaling in MΦs promotes platelet recruitment, we measured MΦs expression of a panel of adhesion molecules known to be important in platelet recruitment (*Hamburger and McEver, 1990*; *Hitchcock et al., 2015*; *Larsen et al., 1989*; *Simon et al., 2000*). Our data showed that podoplanin is expressed at a much higher level in hepatic MΦs isolated from APAP-treated WT mice than those from *Chil1-/-* or *Cd44-/-* mice (*Figure 5A*). Interestingly, rmChi3l1 treatment of *Chil1-/-*, but not *Cd44-/-* mice, markedly increased the podoplanin mRNA and protein expression levels in MΦs (*Figure 5B,C*). To examine the role of podoplanin in mediating platelet adhesion to MΦs, we blocked podoplanin using an anti-podoplanin antibody in *Chil1-/-* mice reconstituted with rmChi3l1. As shown in *Figure 5D–F*, blockade of podoplanin not only abrogated rmChi3l1-mediated platelet recruitment into the liver but also significantly reduced its effect on increasing AILI in *Chil1-/-* mice.

Clec-2 is the only platelet receptor known to bind podoplanin (*Kerrigan et al., 2012*). To further elucidate the role of podoplanin in mediating platelet adhesion to MΦs, we isolated MΦs from WT mice treated with APAP. After treating MΦs with anti-podoplanin antibody or IgG as control, we added platelets. IF staining of podoplanin and Clec-2 showed that the Clec-2-expressing platelets only bound to IgG-treated, but not anti-podoplanin-treated MΦs (*Figure 5—figure supplement 1*). Together, our data demonstrate that MΦs recruit platelets through podoplanin and Clec-2 interaction, and that the podoplanin expression on MΦs is regulated by Chi3l1/CD44 signaling.

## Evaluation of the therapeutic potential of targeting Chi3l1 in the treatment of AILI

Although NAC greatly reduces morbidity and mortality from ALF due to APAP overdose, the death rate and need for liver transplantation remain unacceptably high. While elucidating the underlining biology of Chi3l1 in AILI, we also generated mAbs specifically recognizing either mouse or human Chi3l1. We screened a panel of anti-mouse Chi3l1 monoclonal antibodies (α-mChi3l1 mAb) to determine their efficacies in attenuating AILI. We injected WT mice with an α-mChi3l1 mAb or IgG at 3 hr after APAP challenge. Our data showed that clone 59 (C59) had the most potent effects on inhibiting APAP-induced hepatic platelet accumulation and attenuating AILI (*Figure 6A–C*).

To evaluate the potential of targeting Chi3l1 as a treatment for AILI in humans, we screened all of the α-hChi3l1 mAb we generated by IHC staining of patients' liver biopsies (data not shown) and selected the best clone for in vivo functional studies. Because the amino acid sequence homology between human and mouse Chi3l1 is quite high (76%), we treated *Chil1-/-* mice with rhChi3l1. We found that rhChi3l1 was as effective as rmChi3l1 in promoting platelet recruitment and increasing AILI in *Chil1-/-* mice (*Figure 6D–F*). To our excitement, the α-hChi3l1 mAb treatment could abrogate platelet recruitment and dramatically reduce liver injury (*Figure 6D–F*). Together, these data indicate that mAb-based blocking of Chi3l1 may be an effective therapeutic strategy to treat AILI and potentially other acute liver injuries.

## Discussion

The current study unveiled an important function of Chi3l1 in promoting platelet recruitment into the liver after APAP overdose, thereby playing a critical role in exacerbating APAP-induced

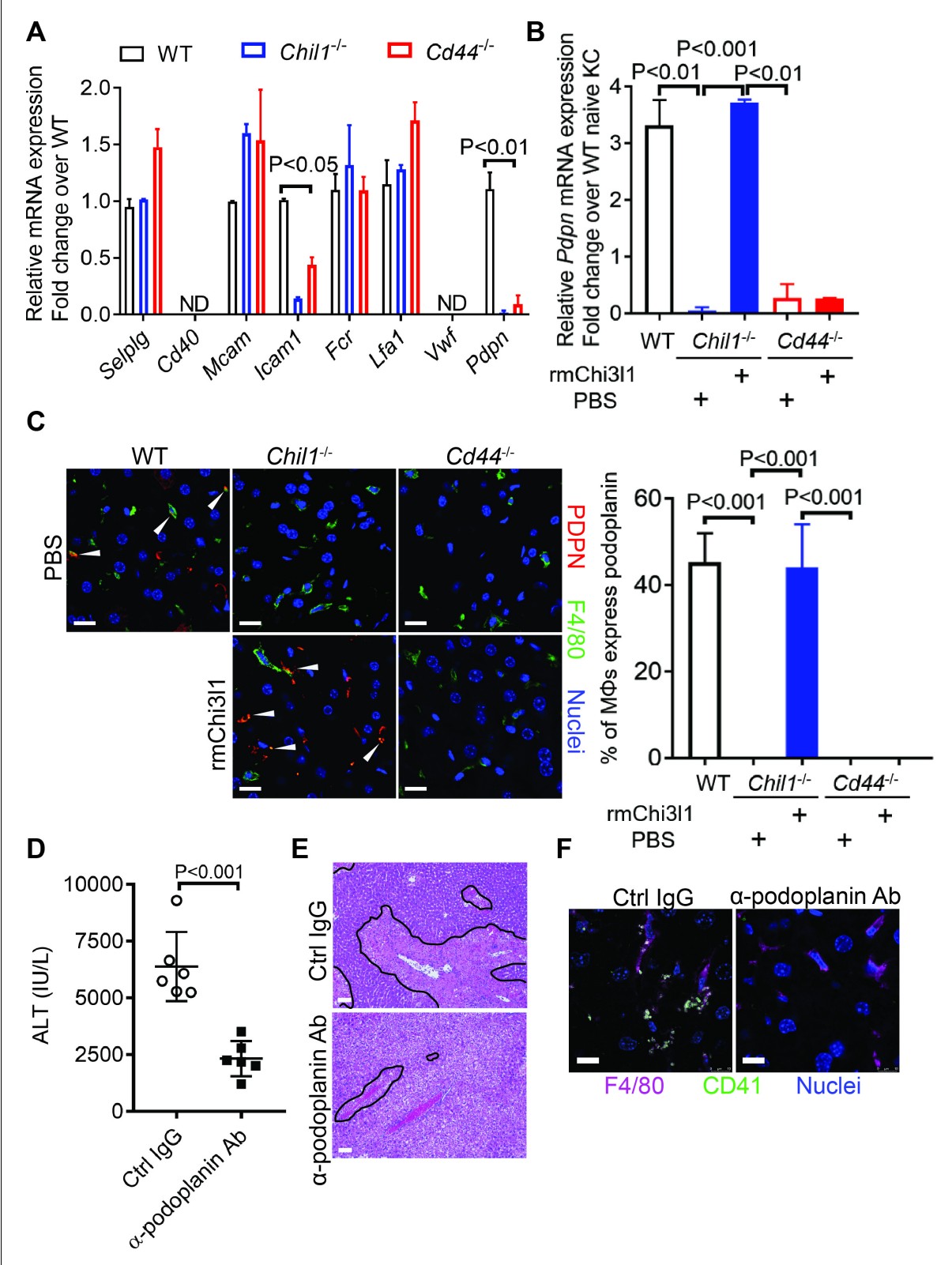

**Figure 5.** Chitinase 3-like-1 (Chi3l1)/CD44 signaling in MΦs upregulates podoplanin expression and platelet adhesion. (**A**) Male WT, *Chil1⁻ᐟ⁻*, *Cd44⁻ᐟ⁻* mice were treated with acetaminophen (APAP) (n = 4 mice/group). After 3 hr, mice were sacrificed and MΦs were isolated to measure mRNA levels of various adhesion molecules, including *selectin P ligand* (*Selplg*), *Cd40*, *melanoma cell adhesion molecule* (*Mcam*), *Fc receptor* (*Fcr*), *intercellular adhesion molecule 1* (*Icam1*), *lymphocyte function-associated antigen 1* (*Lfa1*), *von Willebrand factor* (*Vwf*), and *podoplanin* (*Pdpn*). (**B, C**) Wild-
*Figure 5 continued on next page*

*Figure 5 continued*

type (WT) mice were treated with APAP. *Chil1⁻/⁻* and *Cd44⁻/⁻* mice were treated with PBS or rmChi3l1 followed by APAP challenge simultaneously and mice were sacrificed 3 hr after APAP (n = 3 mice/group). (B) MΦs were isolated and mRNA levels of *Pdpn* in MΦs were analyzed by quantitative reverse transcription polymerase chain reaction (qRT-PCR). (C) Immunofluorescence (IF) staining of liver sections for podoplanin and F4/80 is shown and the proportions of MΦs that express *Pdpn* were quantified, Scale bar, 25 µm. (D–F) *Chil1⁻/⁻* mice reconstituted with rmChi3l1 were treated with either Ctrl IgG or α-podoplanin Ab for 16 hr and subsequently challenged with APAP. (D) Serum levels of ALT and (E) liver histology were evaluated 24 hr after APAP treatment (n = 6 mice/group). Scale bar, 250 µm. (F) IF staining for intrahepatic platelets (CD41⁺) and MΦs (F4/80+) was performed 3 hr after APAP (n = 3 mice/group). Scale bar, 25 µm. One-way ANOVA were performed in A–C. Two-tailed, unpaired Student's t-test was performed in D. The online version of this article includes the following figure supplement(s) for figure 5:

**Figure supplement 1.** Podolanin expressing on MΦs mediates interactions with platelets.

coagulopathy and liver injury. Our data demonstrate that Chi3l1 signals through CD44 on MΦs to

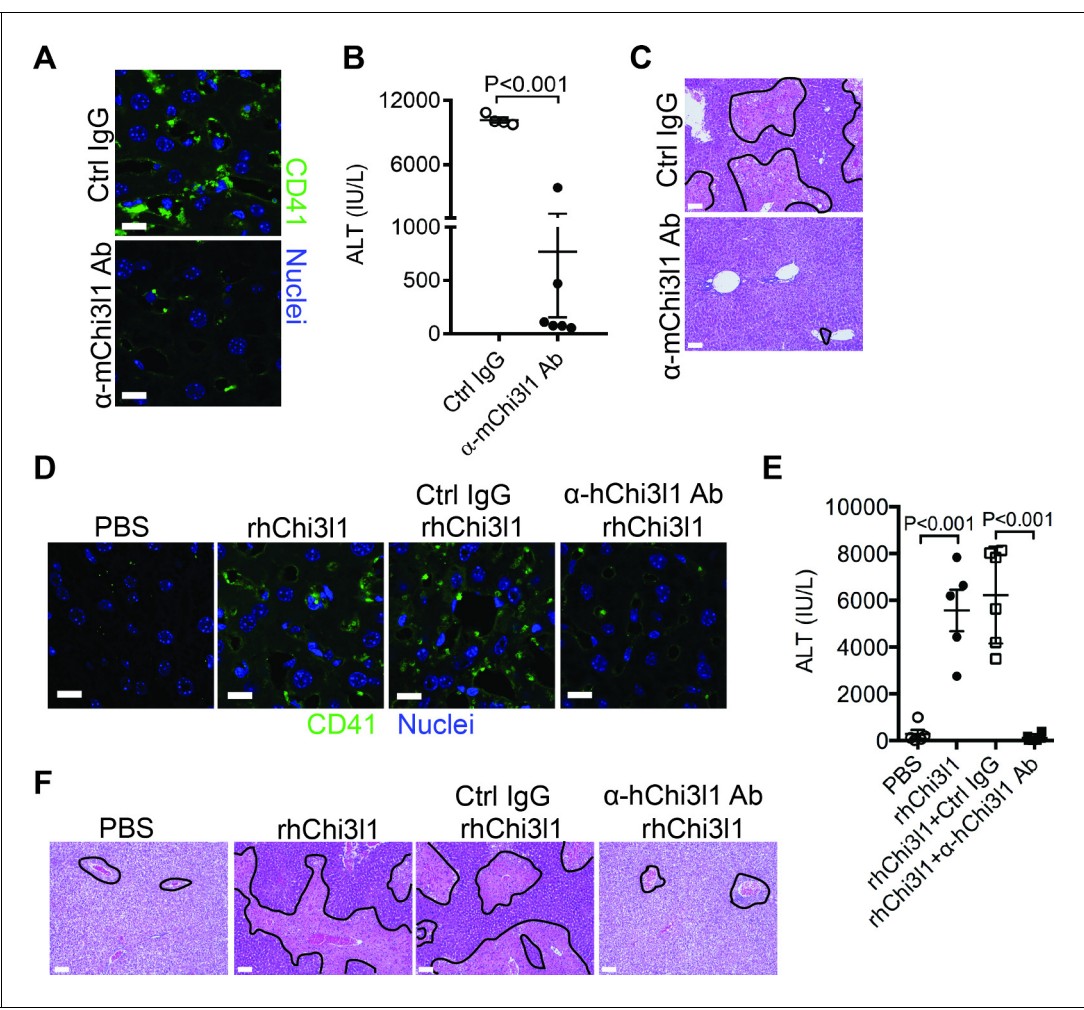

**Figure 6.** Evaluation of the therapeutic potential of targeting chitinase 3-like-1 (Chi3l1) in the treatment of acetaminophen-induced liver injury (AILI). (A–C) Male C57B/6 mice were treated with acetaminophen (APAP) for 3 hr, followed by intraperitoneally (*i.p.*) injection of either a control IgG (Ctrl IgG) or an anti-mouse Chi3l1 Ab (α-mChi3l1 Ab, C59). (A) Immunofluorescence (IF) staining for intrahepatic platelets (CD41⁺) was performed 6 hr after APAP treatment (n = 3 mice/group). Scale bar, 25 µm. (B) Serum levels of ALT and (C) liver histology were evaluated 24 hr after APAP treatment (n = 4–6 mice/group). Scale bar, 250 µm. (D–F) *Chil1⁻/⁻* mice were treated with APAP plus PBS or recombinant human Chi3l1 (rhChi3l1) for 3 hr as indicated and APAP plus rhChi3l1 treatment group were either without treatment or treated with a control IgG (Ctrl IgG) or an anti-human Chi3l1 Ab (α-hChi3l1 Ab, C7). (D) IF staining was performed to identify intrahepatic platelets (CD41⁺) 6 hr after APAP treatment. Scale bar, 25 µm. (E) Serum levels of ALT and (F) liver histology were evaluated 24 hr after APAP treatment. Scale bar, 250 µm (n = 5–10 mice/group in D–F). Two-tailed, unpaired Student's t-test was performed in B. One-way ANOVA were performed in E.

upregulate podoplanin expression and promote platelet recruitment (*Figure 7*). Moreover, we report for the first time significant hepatic accumulation of platelets and marked upregulation of Chi3l1 in patients with ALF caused by APAP overdose. Importantly, we demonstrate that neutralizing Chi3l1 with mAbs can effectively inhibit hepatic platelet accumulation and mitigate liver injury caused by APAP, supporting the potential and feasibility of targeting Chi3l1 as a therapeutic strategy to treat AILI.

The elevation of serum levels of Chi3l1 has been observed in various liver diseases (*Kumagai et al., 2016*; *Lee et al., 2011*; *Nøjgaard et al., 2003*; *Wang et al., 2020*), but studies of its involvement in liver diseases have only begun to emerge. There are several reports describing a role of Chi3l1 in models of chronic liver injuries caused by alcohol, CCl4, or high-fat diet (*Higashiyama et al., 2019*; *Lee et al., 2019*; *Qiu et al., 2018*; *Zhang et al., 2021*). However, the molecular and cellular mechanisms accounting for the involvement of Chi3l1 have yet to be defined. The present study unveils a function of Chi3l1 in promoting platelet recruitment to the liver during acute injury. We provide compelling data demonstrating that Chi3l1 acts through its receptor CD44 on MΦs to recruit platelets, thereby contributing to AILI. Multiple receptors of Chi3l1 have been identified, including IL-13Rα2, CRTH2, TMEM219, and galectin-3 (*Geng et al., 2018*; *He et al., 2013*; *Lee et al., 2016*; *Zhou et al., 2015*; *Zhou et al., 2018*). The fact that Chi3l1 could bind to multiple receptors is consistent with a diverse involvement of Chi3l1 under different disease contexts. A recent study showed that Chi3l1 was upregulated during gastric cancer (GC) development and that through binding to CD44, it activated Erk, Akt, and β-catenin signaling, thereby enhancing GC metastasis (*Geng et al., 2018*). Our studies illustrated a novel role of Chi3l1/CD44 interaction in the recruitment of hepatic platelets and contribution to AILI. Our in vivo studies using $Cd44^{-/-}$ mice and anti-CD44 antibody provide strong evidence that CD44 mediates the effects of Chi3l1. Our observation that Chi3l1 predominantly binds to CD44 on MΦs, but not other CD44-expressing cells in the liver, suggests two possibilities which warrant further investigation. First, Chi3l1 may bind a specific isoform of CD44 that is uniquely expressed by MΦs. Second, the Chi3l1-CD44 interaction

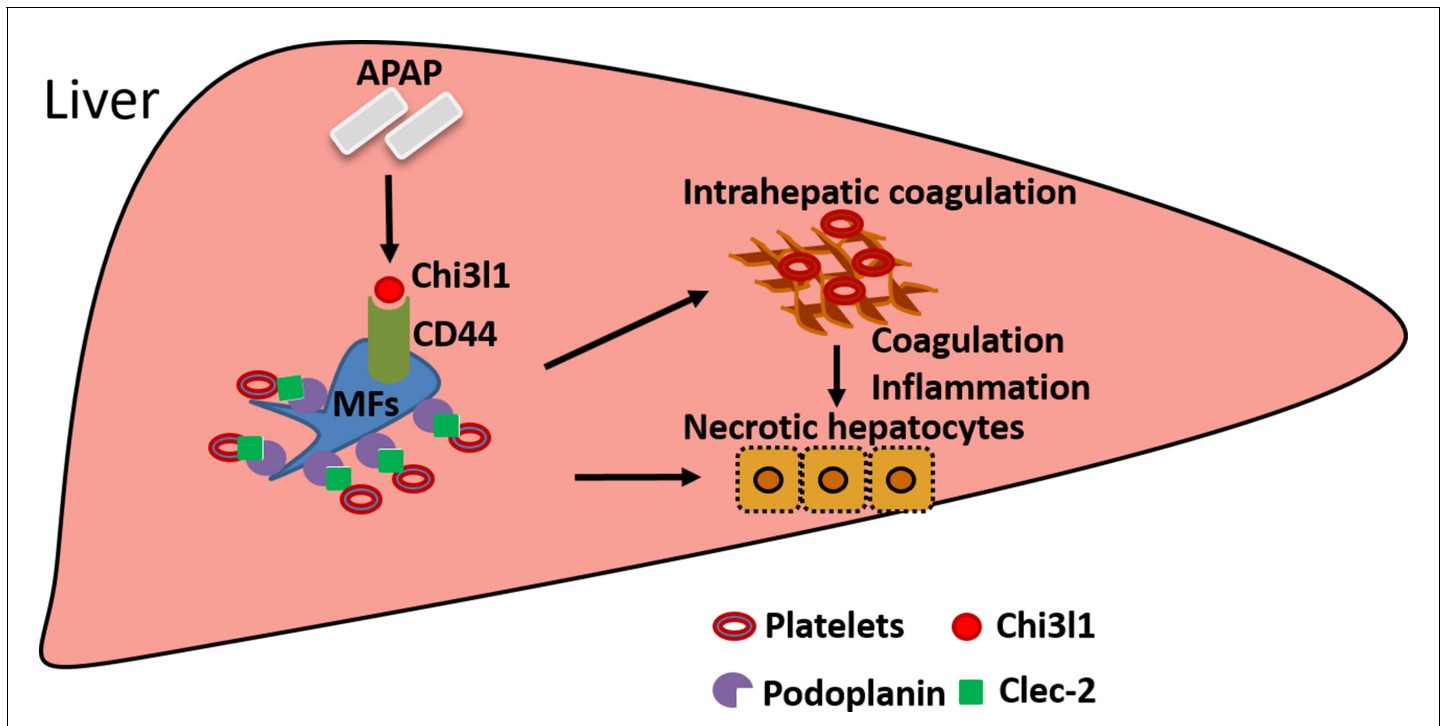

**Figure 7.** Schematic summary of the main findings. Acetaminophen (APAP) overdose induces chitinase 3-like-1 (Chi3l1) expression, which binds CD44 on MΦs and promotes MΦs-mediated platelets recruitment through podoplanin/Clec-2 (C-type lectin-like receptor 2) interaction. Recruited platelets further contribute to APAP-induced liver injury (AILI).

requires binding of a co-receptor, which is expressed on MΦs but not on other CD44-expressing cells in the liver.

We identified hepatic MΦs as a key player in promoting platelet recruitment to the liver during AILI. Given the involvement of platelets in AILI, this finding would suggest that hepatic MΦs also contribute to liver injury. The role of hepatic MΦs in AILI has been a topic of debate and the current understanding is confined by the limitation of the methods used to deplete these cells (*Campion et al., 2008*; *Fisher et al., 2013*; *Ju et al., 2002*; *Laskin et al., 1995*; *Michael et al., 1999*). Several previous studies using CLDN to deplete MΦs concluded that these cells play a protective role against AILI (*Campion et al., 2008*; *Fisher et al., 2013*; *Ju et al., 2002*). However, in those studies, MΦ-depletion was confirmed by IHC staining of F4/80, which cannot distinguish KCs from IMs. Our laboratory and others had since developed a flow cytometric approach to detect and distinguish the two MΦs populations. Using flow cytometry to monitor MΦ-depletion, we found that the timing of CLDN treatment was critical. In the previously published reports, mice were treated with CLDN around 2 days before APAP challenge (*Campion et al., 2008*; *Fisher et al., 2013*; *Ju et al., 2002*). Using this treatment regimen, IMs became abundant prior to APAP treatment, even though KCs were depleted. Without this knowledge, previous studies attributed the worsened AILI to the depletion of KCs. However, the advancement of knowledge on the recruitment of IMs and their contribution to acute liver injury offers an alternative interpretation that the worsened AILI is due to IM accumulation (*Chauhan et al., 2020*; *Holt et al., 2008*; *Mossanen et al., 2016*; *Zigmond et al., 2014*). In the current study, we analyzed KCs and IMs in the liver at various time points after CLDN treatment to identify a new strategy to achieve more complete hepatic MΦ-depletion. Our data demonstrated that when both MΦs populations were absent at the time of APAP treatment, platelet recruitment was abrogated and AILI was significantly reduced. During the preparation of this manuscript, a study was published describing that IMs could recruit platelets (*Chauhan et al., 2020*). Together, these data suggest that hepatic MΦs (both KCs and IMs) play a crucial role in promoting hepatic platelet accumulation, thereby contributing to AILI.

Our data suggest that platelet-derived Clec-2 interacts with podoplanin expressed on MΦs, resulting in platelet recruitment to the liver during the early phase of AILI. The role of podoplanin/Clec-2 interaction in platelet recruitment and thromboinflammation has been indicated in multiple inflammatory and infectious conditions (*Chauhan et al., 2020*; *Hitchcock et al., 2015*; *Kerrigan et al., 2012*). Our data, for the first time, provide evidence that the podoplanin expression on MΦs is regulated by the Chi3l1/CD44 axis. Future studies focusing on gaining molecular insight into such regulation are warranted. An increasing number of studies suggest that platelets play an important, but paradoxical role in liver injury. It has been proposed that they contribute to tissue damage during injury phase but promote tissue repair at later time points (*Chauhan et al., 2016*). However, two recent studies of AILI demonstrate that persistent platelet accumulation in the liver significantly delays liver repair. One study described a podoplanin/Clec-2 interaction between platelets and hepatic IMs during tissue repair and demonstrated a detrimental role of such interaction through blocking the recruitment of reparative neutrophils (*Chauhan et al., 2020*). Another study showed that AILI was associated with elevated plasma levels of von Willebrand factor, which prolonged hepatic platelet accumulation and delayed repair of APAP-injured liver in mice (*Groeneveld et al., 2020*). These studies together with our finding that platelets drive tissue damage during early stage of AILI suggest that platelets may be a therapeutic target to treat acute liver injury.

We observed hepatic platelet accumulation as early as 3 hr after APAP treatment in mice, prior to APAP-induced liver necrosis, indicating that platelets are likely to be the driver of AILI. Mitochondrial damage is a key event in APAP-induced cell necrosis, in which APAP triggers c-jun N-terminal kinase (JNK) activation in the cytosol and translocation of phospho-JNK to the mitochondria, resulting in oxidant stress and the mitochondrial permeability transition pore opening (*Saito et al., 2010*). Others and our lab have reported that Chi3l1 can induce phosphorylation of JNK directly in either bronchial epithelial cells or LSEC line (*Shan et al., 2018*; *Tang et al., 2013*). However, whether Chi3l1 or Chi3l1-recruited platelets affects mitochondrial damage or mitochondrial JNK activation in hepatocytes warrants further investigation. During this study, we did compare the liver injuries among WT, and *Chil1*⁻/⁻, *Cd44*⁻/⁻ mice in the recovery/regeneration stage of AILI (data not shown). Although ALT levels of WT mice were still slightly higher than both knockout strains of mice at 48 hr post-APAP, it is most likely due to high degrees of injury in WT at the initiation stage of AILI but not

due to delayed repair. There were no differences in ALT levels at 72 hr post-APAP, again indicating that the Chi3l1/CD44 does not affect tissue recovery. Moreover, we compared ALT levels at 6 hr post-APAP and there were lower in $Chil1^{-/-}$ and $Cd44^{-/-}$ mice than WT mice (data not shown), which were consistent with the data shown at 24 hr, indicating that Chi3l1/CD44 axis is involved in the initiation and injury phases of AILI.

Our studies uncovered a previously unrecognized involvement of the Chi3l1/CD44 axis in AILI and provided insights into the mechanism by which Chi3l1/CD44 signaling promotes hepatic platelet accumulation and liver injury after APAP challenge. Taking our findings one -step further toward clinical application, we demonstrated the feasibility of targeting Chi3l1 by mAbs to attenuate AILI. There is an unmet need for developing treatments for AILI, as NAC is the only antidote at present. However, the efficacy of NAC declines rapidly when initiated more than a few hours after APAP overdose, long before patients are admitted to the clinic with symptoms of severe liver injury (*Larson et al., 2005*). Our studies provide strong support for the potential targeting of Chi3l1 as a novel therapeutic strategy to improve the clinical outcomes of AILI and perhaps other acute liver injury conditions.

## Acknowledgements

We appreciate the time and effort from Dr Yanyu Wang (Department of Anesthesiology, UTHealth), who diligently double-checked the raw data for each figure.

## Additional information

### Funding

| Funder | Grant reference number | Author |
|---|---|---|
| National Natural Science Foundation of China | 32071129 | Zhao Shan |
| National Institutes of Health | GM123261 | Fong Wilson Lam |
| National Institutes of Health | DK122708 | Cynthia Ju |
| National Institute of Diabetes and Digestive and Kidney Diseases | DK058369 | William M Lee |
| Cancer Prevention and Research Institute of Texas | RP150551 | Zhiqiang An |
| Welch Foundation | AU-0042-20030616 | Zhiqiang An |
| National Institutes of Health | DK109574 | Cynthia Ju |
| National Institutes of Health | DK121330 | Cynthia Ju |
| National Institutes of Health | DK122796 | Cynthia Ju |
| Cancer Prevention and Research Institute of Texas | RP190561 | Zhiqiang An |

The funders had no role in study design, data collection and interpretation, or the decision to submit the work for publication.

### Author contributions

Zhao Shan, Conceptualization, Data curation, Formal analysis, Funding acquisition, Validation, Methodology, Writing - original draft, Project administration, Writing - review and editing; Leike Li, Xun Gui, Data curation, Methodology; Constance Lynn Atkins, Data curation, Validation, Writing - review and editing; Meng Wang, Methodology; Yankai Wen, Jongmin Jeong, Nicolas F Moreno, Writing - review and editing; Dechun Feng, William M Lee, Resources, Writing - review and editing; Ningyan Zhang, Chun Geun Lee, Jack A Elias, Resources; Bin Gao, Zhiqiang An, Conceptualization, Resources, Writing - review and editing; Fong Wilson Lam, Data curation, Investigation, Methodology;

Cynthia Ju, Conceptualization, Resources, Supervision, Funding acquisition, Investigation, Project administration, Writing - review and editing

### Author ORCIDs

Zhao Shan (ID) https://orcid.org/0000-0001-5064-1023
Yankai Wen (ID) http://orcid.org/0000-0002-8144-1515
Zhiqiang An (ID) http://orcid.org/0000-0001-9309-2335
Cynthia Ju (ID) https://orcid.org/0000-0002-1640-7169

### Ethics

Human subjects: Serum samples from patients diagnosed with APAP-induced liver failure on day 1 of admission were obtained from the biobank of the Acute Liver Failure Study Group (ALFSG) at UT Southwestern Medical Center, Dallas, TX, USA. The study was designed and carried out in accordance with the principles of ALFSG and approved by the Ethics Committee of ALFSG (HSC-MC-19-0084). Formalin-fixed, paraffin-embedded human liver biopsies from patients diagnosed with APAP-induced liver failure were obtained from the National Institutes of Health-funded Liver Tissue Cell Distribution System at the University of Minnesota, which was funded by NIH contract # HHSN276201200017C.

Animal experimentation: Animal studies described have been approved by the UTHealth Institutional Animal Care and Use Committee (IACUC AWC-20-0074).

## Additional files

### Supplementary files

• Source data 1. Source data for all numerical bar graph shown in the manuscript including figure supplements.

• Supplementary file 1. Representative list of potential chitinase 3-like-1 (Chi3l1)-interacting proteins detected by mass spectrometry. Non-parenchymal cells were isolated from C57B/6 mice treated with acetaminophen (APAP) for 3 hr and the cell lysate was incubated with rmChi3l1 overnight. Proteins potentially bound to rmChi3l1 were immune-precipitated with an anti-His antibody and subjected to mass spectrometry analyses.

• Transparent reporting form

### Data availability

Intravital microscopy videos can be reached via the following links: https://bcm.box.com/s/15hmtryyrdl302mihrsm034ure87x4ea (Supplemental video 1, PBS treatment) and https://bcm.box.com/s/tuljfmstvv4lvoksx16fkxkpirkekynz (Supplemental Video 2, APAP treatment) (n=6-7 mice/group, 4-15 videos/mouse).

The following datasets were generated:

| Author(s) | Year | Dataset title | Dataset URL | Database and Identifier |
|---|---|---|---|---|
| | 2021 | Supplemental video 1, PBS treatment | | Baylor College of Medicine, 4ea |
| | 2021 | Supplemental Video 2,APAP treatment  tuljfmstvv4lvoksx16fkxkpirkekynz | https://bcm.box.com/s/tuljfmstvv4lvoksx16fkxk-pirkekynz | Baylor College of  Medicine, |

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
