## [Decision Letter]

**Acceptance summary:**

In this manuscript, the authors investigated the role and potential mechanisms of chitinase 3-like-1 (Chi3l1) in hepatic platelet recruitment during acetaminophen-induced liver injury (AILI). They found that human liver samples from APAP overdose and mouse livers treated with APAP had increased Chi3l1 and platelets in the liver. Chil3l1^-/-^ mice were protected against AILI. They further found that Chi3l1 bound with CD44 on macrophages that were critical for platelet recruitment. Moreover, anti-Chi3l1 monoclonal antibodies could effectively inhibit hepatic platelet accumulation and AILI in mice. The findings of chil31 on platelet recruitment and AILI are novel and have high clinical relevance and translational values. This paper will have a high impact as it provided compelling data on the role of macrophages-platelets and novel mechanistic insights in AILI and may give insights into other models of acute liver injury.

**Decision letter after peer review:**

Thank you for submitting your article "Chitinase 3-like-1 Contributes to Acetaminophen-induced Liver Injury by Promoting Hepatic Platelet Recruitment" for consideration by *eLife*. Your article has been reviewed by 3 peer reviewers, and the evaluation has been overseen by Paul Noble as the Senior and Reviewing Editor. The following individual involved in review of your submission has agreed to reveal their identity: Wenxing Ding (Reviewer #1).

Essential revisions:

1) Please review the recommendations of the 3 reviewers and modify the manuscript according to their suggestions. New data are not required if they are not available.

*Reviewer #1:*

In this manuscript, authors investigated the role and potential mechanisms of chitinase 3-like-1 (Chi3l1) in hepatic platelet recruitment during acetaminophen-induced liver injury (AILI). They found that human liver samples from APAP overdose and mouse livers treated with APAP had increased Chi3l1 and platelets in the liver. Chil3l1^-/-^ mice were protective against AILI. They further found that Chi3l1 bound with CD44 on macrophages that were critical for platelet recruitment. Pharmacological depletion of macrophages decreased platelet recruitment and AILI. Moreover, anti-Chi3l1 monoclonal antibodies could effectively inhibit hepatic platelet accumulation and AILI in mice. The manuscript was well written and generally well designed with high quality data. The conclusions were generally supported by the data. The combination of genetic mouse models and antibody manipulations are excellent approaches. The findings of chil31 on platelet recruitment and AILI are novel and have high clinical relevance and translational values. This paper will have a high impact as it provided compelling data on the role of macrophages-platelets and novel mechanistic insights in AILI.

The noticeable limitation was that only a single time point injury of AILI was determined in most experiments. Another limitation is the use of whole body Chi3l1 KO mice and the potential functions/mechanisms of Chi3l1 and platelets in the early phase of cell death induced by APAP, which remains to be determined.

Comments for the authors:

1. Increased hepatic platelets in human liver samples with APAP overdose are most likely reflect very late phase of APAP hepatotoxicity. However, authors observed increased hepatic platelet recruitment/accumulation as early as 3 hours, which was likely prior to the notable necrosis/liver injury. These observations would suggest a potential role of platelet involving in the early phase of APAP-induced necrosis. How would Chi3l1 or platelet affect the well-known pathways that trigger necrosis such as mitochondrial damage and mitochondrial JNK activation by APAP? These questions have not been addressed by the authors. Moreover, the protective effects of posttreatment with anti-Chi3l1 antibody would suggest that it may also involves in the late phase of AILI. These possibilities should be discussed.

2. In almost all the experiments, only one single time point of liver injury after APAP was determined. There are very distinctive phases of AILI (initiation, injury and recovery/regeneration), authors should discuss these limitations in the text. This may also help to clarify some of the conflicting results regarding the role of macrophages and platelets in AILI.

3. Figure 1A, could authors comment on what cell types were the Chi3l1 positive in the patient samples? The staining pattern tends to suggest they are mainly expressed in hepatocytes? Was it possible that Chi3l1 may have other functions in addition to platelet recruitment in AILI? For instance, if Chi3l1 also expresses in hepatocytes would hepatic Chi3l1 have direct effects on hepatocytes in promoting APAP hepatotoxicity? The use of whole body Chi3l1 KO mice is a limitation to address these concerns.

4. Figure 2E the quantification data should also include the WT mice.

5. The hepatotoxicity of APAP is mainly mediated by its metabolite NAPQI. It is critical to rule out any potential changes on the bioactivation of APAP in all the approaches by manipulating Chi3l1. While authors have compared the bioactivation key factors (the levels of hepatic Cyp2e1, early phase of GSH depletion as well as the formation of APAP-adducts in WT and Chi3l1 as well as CD44 KO mice). However, some of the bioactivations were missing in experiments in mice receiving recombinant Chi3l1, anti-Chi3l1 or clodronate for macrophage depletion. Future experiments to dissect the mechanisms of how platelets affect the early phase of hepatocyte necrosis will be exciting although it may be beyond the scope of the current manuscript.

*Reviewer #2:*

Shan et al. investigated mechanism of acetaminophen (APAP)-induced liver injury (AlLI), a life threatening condition with limited therapeutic options. It is known that platelet accumulation contribute to AILI, but its mechanism remains fully elucidated. A previous study by the authors showed a role of chitinase 3-like 1 (Chi3l1) in hepatic coagulation, linking its relation to the platelets. In the current study, the authors first demonstrated that Chi3l1 promotes AILI. Second, they showed that hepatic accumulation of platelets in mice promotes AILI. Third, they showed Chi3I1 is important for platelet accumulation in AILI. To determine the mechanistic link between Chi3I1 and platelet accumulation in the liver, the authors identified its receptor, CD44, by incubating non-parenchymal cells with His-tagged recombinant mouse (rm) Chi3I1 and pull-downed by immunoprecipitation against His-tag, followed by the LC/MS. Vigorous in vivo and in vitro kinetic experiments confirmed a direct interaction between Chi3I1 and CD44. Interestingly, CD44 expressing cells were identified to be macrophages. Using adhesion molecule panel, it was found that podoplanin in macrophages (later found to be induced by Chi3I1) mediated platelet adhesion to macrophages. Further, the authors identified that C-type lectin-like receptor 2 (Clec-2) on platelets binds to podoplanin on macrophages, leading to the macrophage recruitment of platelets. Finally, authors evaluated therapeutic potential of Chi3I1 for the treatment of AILI using an anti-mouse Chi3I1 and showed that anti-mChi3I1 could block APAP-induced platelet recruitment and liver injury in mice.

Elegant and vigorous experiments were performed to demonstrate the role of Chi3I1 in the APAP-induced liver injury. This study established a novel concept that hepatic macrophages as a key player in promoting platelet recruitment to the liver in AILI. The finding is significant because there is only limited treatment option for AILI and this study demonstrated that Chi3I1 can be a novel target for the treatment of AILI.

*Reviewer #3:*The principal involvement of platelets in acetaminophen-induced liver injury is well known. Cynthia Ju's group now provides a very comprehensive and conclusive experimental study, implicating Chi3l1 signaled through CD44 on macrophages to induce podoplanin expression, which mediated platelet recruitment through C-type lectin-like receptor 2.1. YKL-40 is generally known as a fibrosis marker in humans. In Figure 1, it would be interesting to understand the range of YKL-40 seen ALF as compared to liver fibrosis ("diseased control group").

2. In a similar direction, do you have data from human patients correlating serum YKL-40 with the extent of liver injury (e.g. peak ALT / bilirubin / INR) and prognosis (recovery vs. transplant vs. death)? Do YKL-40 levels correlate (inversely) with platelet count?

3. I found the experimental data very convincing and conclusive. Nonetheless, it is not clear to me why platelets aggravate APAP. Can the authors speculate and/or provide data? Does platelet binding affect the polarization or function of the hepatic macrophages?

4. I share the author's excitement about the α-hChi3l1 mAb approach. From a clinical perspective, it would be interesting to combine this treatment with N-acetyl cysteine treatment. Moreover, it would be important and relevant to see, how "late" you can apply this. 3 hrs after APAP is still early (way before the ALT peak). Would 6h or 12h after APAP still protect from AILI?

---

## [Author Response]

Reviewer #1:In this manuscript, authors investigated the role and potential mechanisms of chitinase 3-like-1 (Chi3l1) in hepatic platelet recruitment during acetaminophen-induced liver injury (AILI). They found that human liver samples from APAP overdose and mouse livers treated with APAP had increased Chi3l1 and platelets in the liver. Chil3l1^-/-^ mice were protective against AILI. They further found that Chi3l1 bound with CD44 on macrophages that were critical for platelet recruitment. Pharmacological depletion of macrophages decreased platelet recruitment and AILI. Moreover, anti-Chi3l1 monoclonal antibodies could effectively inhibit hepatic platelet accumulation and AILI in mice. The manuscript was well written and generally well designed with high quality data. The conclusions were generally supported by the data. The combination of genetic mouse models and antibody manipulations are excellent approaches. The findings of chil31 on platelet recruitment and AILI are novel and have high clinical relevance and translational values. This paper will have a high impact as it provided compelling data on the role of macrophages-platelets and novel mechanistic insights in AILI.The noticeable limitation was that only a single time point injury of AILI was determined in most experiments. Another limitation is the use of whole body Chi3l1 KO mice and the potential functions/mechanisms of Chi3l1 and platelets in the early phase of cell death induced by APAP, which remains to be determined.Comments for the authors:1. Increased hepatic platelets in human liver samples with APAP overdose are most likely reflect very late phase of APAP hepatotoxicity. However, authors observed increased hepatic platelet recruitment/accumulation as early as 3 hours, which was likely prior to the notable necrosis/liver injury. These observations would suggest a potential role of platelet involving in the early phase of APAP-induced necrosis. How would Chi3l1 or platelet affect the well-known pathways that trigger necrosis such as mitochondrial damage and mitochondrial JNK activation by APAP? These questions have not been addressed by the authors. Moreover, the protective effects of posttreatment with anti-Chi3l1 antibody would suggest that it may also involves in the late phase of AILI. These possibilities should be discussed.

We appreciate reviewer’s suggestions and included further discussion on Page 19-20, Line 505-522.

2. In almost all the experiments, only one single time point of liver injury after APAP was determined. There are very distinctive phases of AILI (initiation, injury and recovery/regeneration), authors should discuss these limitations in the text. This may also help to clarify some of the conflicting results regarding the role of macrophages and platelets in AILI.

We appreciate reviewer’s comments. During this study, we did compare the liver injuries among WT, and *Chil1^-/-^, Cd44^-/-^* mice in the recovery/regeneration stage of AILI (Author response image 1). Although ALT levels of WT mice were still slightly higher than both KO mice at 48hrs post-APAP, we think it is most likely due to high degrees of injury in WT mice at the initiation stage of AILI but not due to delayed repair. Besides, there were no differences in ALT levels at 72hrs post APAP, indicating that the Chi3l1/CD44 does not affect tissue recovery. Besides, we also examined ALT levels at 6hrs post-APAP (Figure 1). The results were consistent with the data shown at 24hrs, indicating that Chi3l1/CD44 axis is involved in the initiation and injury phases of AILI. In the revised manuscript, we included additional discussion regarding this comment on Page 19-20, Line 505-522.

**Author response image 1. respfig1:** Deletion of Chi3l1- or CD44 does not affect liver recovery after APAP-induced injury. Male WT, *Chil1^-^*^/-^ and *Cd44*^-/-^ mice were treated with APAP. Serum ALT levels were measured at 6hrs, 24hrs, 48hrs and 72hrs after APAP treatment (n=5-8 mice/group). One-way ANOVA was performed.

3. Figure 1A, could authors comment on what cell types were the Chi3l1 positive in the patient samples? The staining pattern tends to suggest they are mainly expressed in hepatocytes? Was it possible that Chi3l1 may have other functions in addition to platelet recruitment in AILI? For instance, if Chi3l1 also expresses in hepatocytes would hepatic Chi3l1 have direct effects on hepatocytes in promoting APAP hepatotoxicity? The use of whole body Chi3l1 KO mice is a limitation to address these concerns.

We agree with the reviewer that Figure 1A would suggest that both hepatocytes and macrophages are positive for Chi3l1. Although our data demonstrate that Chi3l1 binds to Kupffer cells and that hepatocytes do not express CD44, we cannot exclude the possibility that Chi3l1 might still act on hepatocytes, perhaps through another receptor. Additional studies of treating hepatocytes with recombinant Chi3l1 in vitro might help to address whether Chi3l1 has a direct effect on hepatocytes in promoting APAP hepatotoxicity.

4. Figure 2E the quantification data should also include the WT mice.

We appreciate the reviewer’s suggestion and included the quantification data of WT mice now in the revised Figure 2E.

5. The hepatotoxicity of APAP is mainly mediated by its metabolite NAPQI. It is critical to rule out any potential changes on the bioactivation of APAP in all the approaches by manipulating Chi3l1. While authors have compared the bioactivation key factors (the levels of hepatic Cyp2e1, early phase of GSH depletion as well as the formation of APAP-adducts in WT and Chi3l1 as well as CD44 KO mice). However, some of the bioactivations were missing in experiments in mice receiving recombinant Chi3l1, anti-Chi3l1 or clodronate for macrophage depletion. Future experiments to dissect the mechanisms of how platelets affect the early phase of hepatocyte necrosis will be exciting although it may be beyond the scope of the current manuscript.

To ensure that gene deletion of Chi3l1 or CD44 did not affect APAP bioactivation, we measured key determining factors. We did not measure these factors in experiments of rChi3l1 or anti-Chi3l1 treatment because these treatments were performed at the same time or 3hrs after APAP challenge, at which time APAP bioactivation has already occurred. Regarding macrophage depletion by clodronate, we and other had shown that this treatment does not affect APAP bioactivation.[1, 2]

We share the excitement with the reviewer and agree that the next question to address would be how platelets affect hepatocyte necrosis during the early phase of injury.

Reviewer #3 (Recommendations for the authors):The principal involvement of platelets in acetaminophen-induced liver injury is well known. Cynthia Ju's group now provides a very comprehensive and conclusive experimental study, implicating Chi3l1 signaled through CD44 on macrophages to induce podoplanin expression, which mediated platelet recruitment through C-type lectin-like receptor 2.1. YKL-40 is generally known as a fibrosis marker in humans. In Figure 1, it would be interesting to understand the range of YKL-40 seen ALF as compared to liver fibrosis ("diseased control group").

We appreciate the reviewer’s suggestions. We agree with the reviewer that YKL-40 is known as a fibrosis marker in humans. Unfortunately, we do not have serum samples from patients with liver fibrosis. However, we compared the levels of YKL40 we detected in AILI patients with published data [3, 4]. The levels of YKL-40 in AILI patients (150-1300 ng/ml) are higher than those in liver fibrosis patients (80-420 ng/ml). This is likely because liver fibrosis is a chronic liver disease while AILI is an acute liver injury, in which hepatocytes undergo necrosis to a greater extend.

2. In a similar direction, do you have data from human patients correlating serum YKL-40 with the extent of liver injury (e.g. peak ALT / bilirubin / INR) and prognosis (recovery vs. transplant vs. death)? Do YKL-40 levels correlate (inversely) with platelet count?

This is a great question that we are also curious about. However, we could only obtain serum samples from patients undergoing transplantation due to APAP-induced liver failure, and only 1 to 2 days before surgery. This limited our ability to correlate YKL-40 levels with the extent of liver injury or with platelet counts. We are pursuing a prospective study to address the exact questions.

3. I found the experimental data very convincing and conclusive. Nonetheless, it is not clear to me why platelets aggravate APAP. Can the authors speculate and/or provide data? Does platelet binding affect the polarization or function of the hepatic macrophages?

In the current study, we did not elucidate the underlying mechanism by which platelets exacerbate APAP-induced liver injury. This is clearly an important question and has not been addressed. There are two studies of the role of platelets in liver repair after APAP-induced injury. One report described a detrimental role of platelets through blocking the recruitment of reparative neutrophils.[5] Another study showed that AILI was associated with elevated plasma levels of von Willebrand Factor (vWF), which prolonged hepatic platelet accumulation and delayed repair of APAP-injured liver in mice.[6] We included additional discussion in the revised manuscript regarding how platelets might aggravate APAP(Manuscript Page 19-20, Line 505-522). So far, we have not found any studies regarding whether platelets could affect the polarization or function of the hepatic macrophages. This idea certainly warrants further investigation.

4. I share the author's excitement about the α-hChi3l1 mAb approach. From a clinical perspective, it would be interesting to combine this treatment with N-acetyl cysteine treatment. Moreover, it would be important and relevant to see, how "late" you can apply this. 3 hrs after APAP is still early (way before the ALT peak). Would 6h or 12h after APAP still protect from AILI?

These are very important questions to address. We have started to evaluate the efficacies of a panel of our antibodies, and the experiment design includes varying the dose and timing of antibody treatments as well as in combination with NAC.

References:

[1] S. N. Campion et al., "Hepatic Mrp4 induction following acetaminophen exposure is dependent on Kupffer cell function," (in English), American Journal of Physiology-Gastrointestinal and Liver Physiology, vol. 295, no. 2, pp. G294-G304, Aug 2008.

[2] C. Ju et al., "Protective role of Kupffer cells in acetaminophen-induced hepatic injury in mice," Chem Res Toxicol, vol. 15, no. 12, pp. 1504-13, Dec 2002.

[3] X. Jin et al., "Serum chitinase-3-like protein 1 is a biomarker of liver fibrosis in patients with chronic hepatitis B in China," (in English), Hepatobiliary and Pancreatic Diseases International, vol. 19, no. 4, pp. 384-389, Aug 2020.

[4] C. Nojgaard et al., "Serum levels of YKL-40 and PIIINP as prognostic markers in patients with alcoholic liver disease," (in English), Journal of Hepatology, vol. 39, no. 2, pp. 179-186, Aug 2003.

[5] A. Chauhan et al., "The platelet receptor CLEC-2 blocks neutrophil mediated hepatic recovery in acetaminophen induced acute liver failure," Nat Commun, vol. 11, no. 1, p. 1939, Apr 22 2020.

[6] D. Groeneveld et al., "Von Willebrand factor delays liver repair after acetaminophen-induced acute liver injury in mice," J Hepatol, vol. 72, no. 1, pp. 146-155, Jan 2020.